# Cx43-mediated hyphal folding counteracts phagosome integrity loss during fungal infection

Beatriz Cristovao,[1,2] Lisa Rodrigues,[3] Steve Catarino,[1,2] Monica Abreu,[1,2] Teresa Gonçalves,[2,3] Neuza Domingues,[1,2] Henrique Girao[1,2]

**ABSTRACT** Phagolysosomes are crucial organelles during the elimination of pathogens by host cells. The maintenance of their membrane integrity is vital during stressful conditions, such as during *Candida albicans* infection. As the fungal hyphae grow, the phagolysosome membrane expands to ensure that the growing fungus remains entrapped. Additionally, actin structures surrounding the hyphae-containing phagosome were recently described to damage and constrain these pathogens inside the host vacuoles by inducing their folding. However, the molecular mechanism involved in the phagosome membrane adaptation during this extreme expansion process is still unclear. The main goal of this study was to unveil the interplay between phagosomal membrane integrity and folding capacity of *C. albicans*-infected macrophages. We show that components of the repair machinery are gradually recruited to the expanding phagolysosomal membrane and that their inhibition diminishes macrophage folding capacity. Through an analysis of an RNAseq data set of *C. albicans*-infected macrophages, we identified Cx43, a gap junction protein, as a putative player involved in the interplay between lysosomal homeostasis and actin-related processes. Our findings further reveal that Cx43 is recruited to expand phagosomes and potentiates the hyphal folding capacity of macrophages, promoting their survival. Additionally, we reveal that Cx43 can act as an anchor for complexes involved in Arp2-mediated actin nucleation during the assembly of actin rings around hyphae-containing phagosomes. Overall, this work brings new insights on the mechanisms by which macrophages cope with *C. albicans* infection ascribing to Cx43 a new noncanonical regulatory role in phagosome dynamics during pathogen phagocytosis.

**IMPORTANCE** Invasive candidiasis is a life-threatening fungal infection that can become increasingly resistant to treatment. Thus, strategies to improve immune system efficiency, such as the macrophage response during the clearance of the fungal infection, are crucial to ameliorate the current therapies. Engulfed *Candida albicans*, one of the most common *Candida* species, is able to quickly transit from yeast-to-hypha form, which can elicit a phagosomal membrane injury and ultimately lead to macrophage death. Here, we extend the understanding of phagosome membrane homeostasis during the hypha expansion and folding process. We found that loss of phagosomal membrane integrity decreases the capacity of macrophages to fold the hyphae. Furthermore, through a bioinformatic analysis, we reveal a new window of opportunities to disclose the mechanisms underlying the hyphal constraining process. We identified Cx43 as a new weapon in the armamentarium to tackle infection by potentiating hyphal folding and promoting macrophage survival.

**KEYWORDS** *Candida albicans*, infectious disease, connexin 43, phagosomes, macrophages

Address correspondence to Henrique Girao, hmgirao@fmed.uc.pt, or Neuza Domingues, neuzadomingues16@gmail.com.

The authors declare no conflict of interest.

See the funding table on p. 17.

Invasive candidiasis, a life-threatening fungal infection, can be caused by several *Candida* spp., *C. albicans* being the most prevalent species worldwide (1, 2). *C. albicans* is a constituent of the microbiota in most healthy individuals (3). However, as opportunistic pathogens, under host immunity weakening, these fungi switch to virulent phenotypes, which can consequently lead to fulminant sepsis (3). Considering that innate immunity is the first line of defense against pathogens (4, 5), a better understanding of the immune cell response to counteract the transition between morphological stages of this microorganism is of utmost importance to design new strategies to strengthen immune system efficiency.

Macrophages recognize, engulf, and eliminate *C. albicans* through phagocytosis (5, 6). After *C. albicans* internalization, the yeast may transit to the hyphal form, which involves the continuous length growth of the engulfed pathogen, which can induce phagosomal membrane damage (7, 8). Thus, to cope with these adverse conditions, macrophages mount an efficient cellular response to rapidly neutralize the hypha-induced cellular damage, avoiding cell death. Until now, the main described strategies adopted by macrophages to kill *C. albicans* encompass an increase in reactive oxygen species and reactive nitrogen species production, a reduction of nutrient availability in the phagolysosome, and a more acidic and hydrolytic environment in the proteolytic compartment (8–13). Nevertheless, as a result of the generated tension from the expanding pathogens, phagolysosomal membrane integrity is commonly compromised, reducing host cell viability (8). Thus, to counteract this deleterious effect, cells have developed strategies to repair and maintain membrane integrity (14–16). These repair responses are triggered by the efflux of phagosomal calcium ($Ca^{2+}$) to the cytosol, with a rapid recruitment of the endosomal sorting complex required for transport (ESCRT) machinery (16, 17) and scramblases to the injured sites (18). Interestingly, some of these repair-related complexes such as ESCRT-I complex, including tumor susceptibility gene 101 protein (Tsg101) (19), and ALG-2-interacting protein X (Alix) (20) were already shown to be enriched in phagosomes. Moreover, these phagolysosomal membrane repair mechanisms have already been associated with cellular response to lysosomal damage induced by pathogens such as *Mycobacterium tuberculosis* (21, 22), *Listeria monocytogenes* (23), and *Salmonella enterica* (18). In the case of macrophage infection with *C. albicans*, it was reported that upon $Ca^{2+}$ signaling, activation of the leucine-rich repeat kinase 2 (LRRK2) (23, 24) promotes the recruitment of the Rab8A and the ESCRT-III component CHMP4B to the yeast-containing phagosome. $Ca^{2+}$ efflux from ruptured membranes was also positively correlated with the activation of lysosomal fusion required to sustain the expansion of the phagolysosome upon hyphae growth (25). Besides lysosomal membrane repair, ESCRT machinery was demonstrated to be required to fix the plasma membrane of epithelial cells infected with *C. albicans* (26). However, the recruitment of ESCRT complex proteins to the phagolysosome containing the expanding hyphae remains unclear. In addition to the membrane repair mechanisms, the assembly of actin structures to fold the expanding hyphae was recently described as an additional defense mechanism used by infected macrophages (27–32). In this process, macrophages promote the assembly of actin ring structures localized around the hypha-containing phagosomes, hampering pathogen growth and escape, and facilitating its complete engulfment by the macrophage phagosome (31). Nevertheless, it is not known how the loss of phagolysosome membrane repair capacity affects the newly described hyphal folding process. Thus, here, we propose to investigate the impact of phagolysosomal membrane integrity on the hyphal folding mechanism and unveil new molecular players in this process. Our data suggest that the recruitment of membrane repair machinery to hyphae-containing phagolysosome is important to potentiate the folding process in macrophages. We also report that Cx43, a gap junction protein, is recruited to expanding phagolysosomes, localizes to actin ring structures, and promotes hyphae folding, through a mechanism that requires Arp2-actin nucleation.

## RESULTS

### Internalized *C. albicans* causes the recruitment of ESCRT machinery components to the expanding phagosomal membrane

Professional phagocytes, such as macrophages, are subjected to extreme adverse conditions when invading organisms trigger abrupt morphological and structural changes of the expanding phagosome that can result in its membrane disruption. In order to restrain the growing pathogen and prevent phagocytic compartment rupture, macrophages can activate fast cellular response to seal the damaged membranes. Aiming to unveil the molecular partners implicated in this process, we evaluated the recruitment of two ESCRT machinery components to the membrane of expanding phagosomes containing *C. albicans*, namely, Alix and Tsg101. To address this question, the macrophage cell line RAW264.7 (RAW cells) was infected with *C. albicans* for up to 180 min and then immunostained for Alix and Tsg101, to assess their recruitment to *C. albicans* containing compartments. Lamp1 was used as a marker of late endosomes and lysosomes. Interestingly, the *C. albicans* yeast-to-hypha transition (180 min) leads to a total increase of Alix and Tsg101 in phagolysosomes containing hyphae (Fig. 1a and b). These findings were further supported by quantitative imaging analysis correlating the length of the internalized *C. albicans* with the relative amount of Alix or Tsg101 at phagolysosomes (Fig. 1c and d), where a positive correlation was found between pathogen length and the extent of these proteins recruitment ($r^2 = 0.5492$ and $r^2 = 0.5889$, respectively). To confirm these data, infected macrophages were treated with 1,2-bis(2-aminophenoxy)ethane-N,N,N′,N′-tetraacetic acid tetrakis(acetoxymethyl ester) (BAPTA-AM), a cytosolic $Ca^{2+}$ chelator, previously shown to inhibit $Ca^{2+}$-dependent recruitment of ESCRT machinery (16). Not surprisingly, the presence of BAPTA-AM decreased the recruitment of both ESCRT components (Fig. S1a through d). Additionally, a more extensive colocalization between Alix or Tsg101 and Lamp1 was observed when phagosomal rupture was further potentiated by L-leucyl-L-leucine methyl ester (LLOMe) (Fig. S1a through b). Of note, total protein levels of these ESCRT components were not significantly affected by the presence of the pathogen (data not shown). We also observed that infection with heat-killed *C. albicans* or induction of phagolysosomal rupture by LLOMe also had no effect on the total expression levels of Alix or Tsg101. Overall, these results strongly suggest that hyphal elongation is accompanied by an increase in ESCRT component recruitment to the phagosome, likely contributing to a continuous maintenance of phagolysosomal membrane integrity.

### Expanding phagosomes do not present lysophagy markers

When the extent and severity of the lysosomal membrane damage overcome or delay repair capacity, the cells can trigger the clearance of these injured organelles through lysophagy, a type of selective autophagy (16). This process is initiated by the recruitment of cytoplasmic galectins and glycoprotein-specific ubiquitin ligases to abnormally exposed luminal glycans at the damaged site, resulting in the engulfment of the injured phagolysosome by autophagic membranes (21, 33). We next sought to elucidate whether lysophagy machinery could be elicited in the cellular response to the growing hypha as a mechanism to provide membrane elements to the expanding phagosome. For this, we assessed the recruitment to phagolysosomes of galectin 3 (Gal3), tripartite motif containing 16 (Trim16), a Gal3 effector, and ubiquitin, previously reported to be implicated in lysosome removal upon damage (21). In hypha-containing phagosomes, Gal3 was slightly recruited to the phagolysosomes (Fig. 2a), and the correlation between Gal3 found in phagosomes containing the hypha and the hypha length was only moderately positive (Fig. 2b), with a coefficient of determination of 0.3614. Moreover, Trim16 recruitment was more evident in hypha-containing phagosomes than in compartments containing the yeast form (Fig. 2c). However, as for Gal3, only a slight positive correlation was observed between Trim16 recruitment to the phagosome and hypha length ($r^2 = 0.3327$) (Fig. 2d). Interestingly, when cells were treated with LLOMe, no

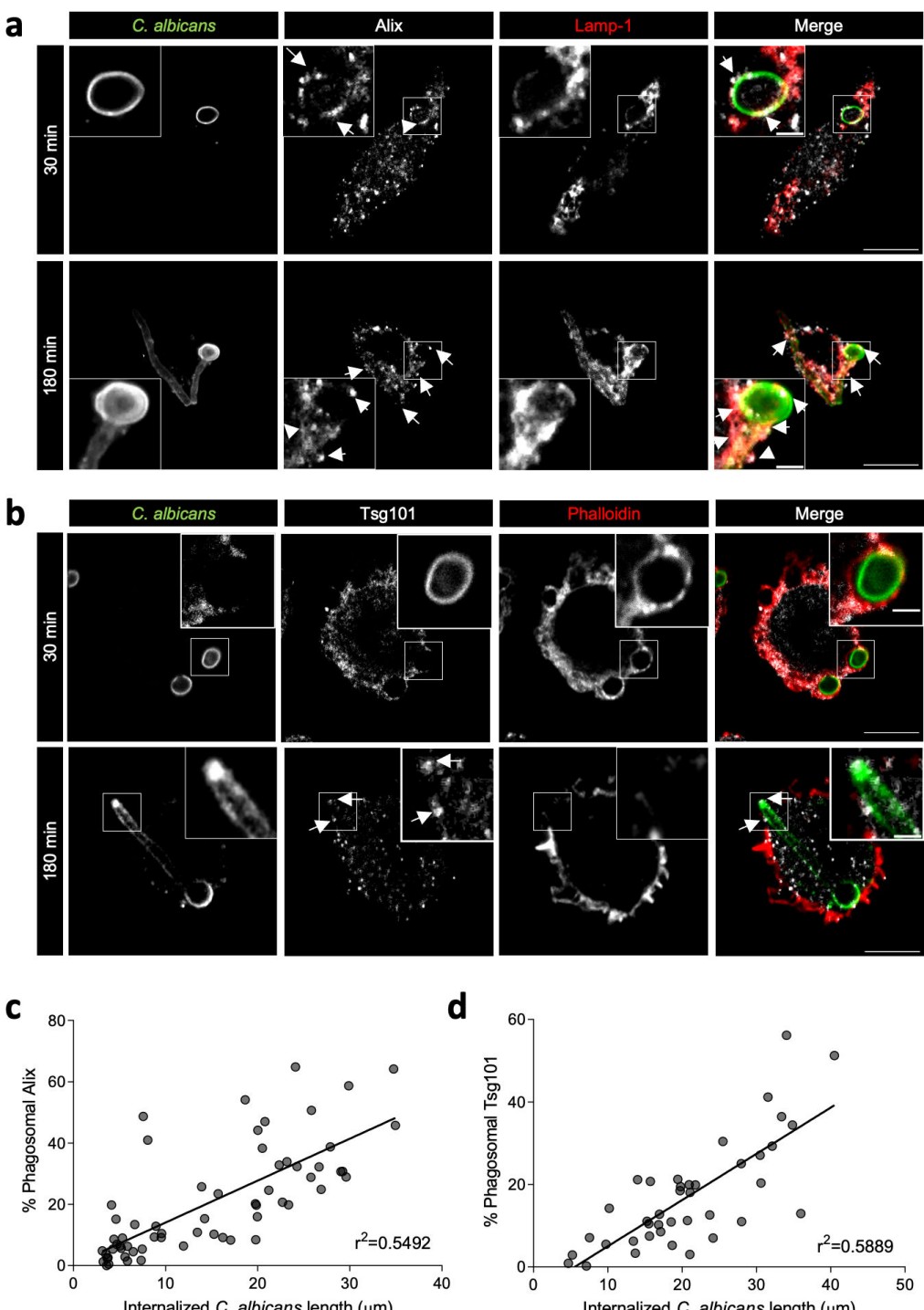

**FIG 1** Internalized *C. albicans* causes the recruitment of ESCRT machinery components to the expanding phagosomal membrane. (a and b) Representative confocal images of RAW cells infected with *C. albicans*, fixed after 30 min and 180 min post-infection, and immunostained for *C. albicans*, ALG-2-interacting protein X (Alix) (a), tumor susceptibility gene 101 protein (Tsg101) (b), lamp-1 and phalloidin. Scale bar, 10 µm and 2 µm in the inset images. White arrows indicate sites of Alix and Tsg101 accumulation. (c) Regression analysis comparing the length of internalized *C. albicans* and the percentage of phagosomal Alix quantified as (phagosomal Alix/total Alix)*100 per each analyzed cell. (d) Regression analysis comparing the length of internalized *C. albicans* and percentage of phagosomal Tsg101 quantified as (phagosomal Tsg101/total Tsg101)*100 per each analyzed cell. Results from three independent experiences.

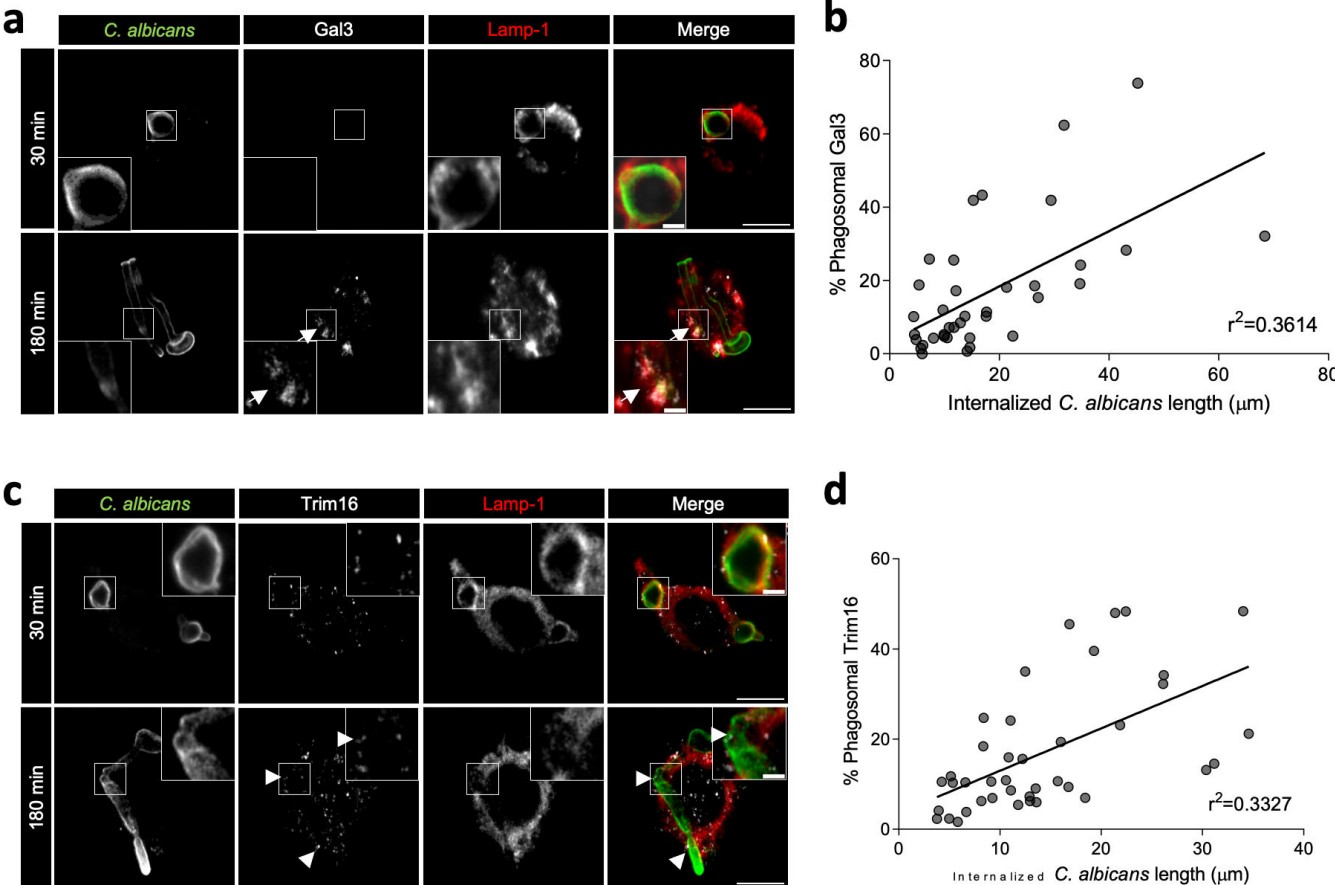

**FIG 2** Expanding phagosomes do not present lysophagy markers. (a) Representative confocal images of RAW cells infected with *C. albicans*, fixed 30 min and 180 min post-infection, and immunostained for *C. albicans*, galectin 3 (Gal3), and Lamp-1. Scale bar, 10 µm and 2 µm in the inset images. White arrows indicate sites of Gal3 accumulation. (b) Regression analysis comparing the length of internalized *C. albicans* and the percentage of phagosomal Gal3 quantified as (phagosomal Gal3/total Gal3)*100 per each analyzed cell. Results from three independent experiments. (c) Representative confocal images of RAW cells infected with *C. albicans*, fixed 30 min and 180 min post-infection, and immunostained for *C. albicans*, tripartite motif containing 16 (Trim16), and Lamp-1. Scale bar, 10 µm and 2 µm, in the inset images. White arrowheads indicate sites of Trim16 accumulation. (d) Regression analysis comparing the length of internalized *C. albicans* and the percentage of phagosomal Trim16 quantified as (phagosomal Trim16/total Trim16)*100 per each infected cell. Results from three independent experiences.

additional recruitment of Gal3 and Trim16 to hypha-containing phagosomes was detected, but it substantially increased the recruitment of Gal3 and Trim16 to other Lamp1-positive vesicles devoid of pathogens (Fig. S2a and b). The total levels of Gal3 and Trim16 did not significantly vary after *C. albicans* infection (data not shown). Similar to Trim16 and Gal3, we also observed the recruitment of ubiquitin to hyphae-containing phagosomes (Fig. S2c), although with only a small correlation between hyphae length and phagosomal ubiquitin fraction (Fig. S2d). Moreover, to disclose the impact of *C. albicans* in autophagy, the total levels of autophagic proteins, such as Beclin-1, p62, and LC3, were assessed by Western blot. The results showed that the levels of these proteins are not significantly altered in macrophages infected with *C. albicans* (Fig. S3). Together, these results provide evidence that lysophagy machinery is not implicated in the cellular response to growing phagolysomes or in maintaining their integrity.

## Phagosomal membrane integrity impacts *C. albicans* hyphal folding

Grounded on recent studies showing that inhibition of the lysosomal fusion reduces the integrity of the phagosome and facilitates hyphal growth (25), we hypothesized that the hyphal folding machinery can be affected by phagosome membrane dynamics, with a

consequent implication in hyphal growth. To evaluate whether defects on phagosomal membrane integrity and autophagy could also affect the folding process, we started by assessing the impact of BAPTA-AM, 3-methyladenine (3-MA, autophagy inhibitor), and LLOMe in internalized hyphae length. In agreement with previous findings (25), we found an increase in hypha size upon treatment with BAPTA-AM, as well as in infected cells treated with 3-MA and LLOMe (Fig. 3a), suggesting that damaged phagosomes are more permissive to hyphal growth. Next, we investigated the effect of phagosomal membrane homeostasis on the hyphal folding. We quantified and categorized the percentage of hyphal folding on *C. albicans* when treated with the aforementioned compounds as none (no visible folding), low folding (>90° angle), or high folding (<90° angle) (Fig. 3b and c). The results showed that loss of phagosomal integrity induced by BAPTA-AM, 3-MA, and LLOMe treatment significantly reduced macrophage capacity to induce hyphal folding (Fig. 3c; Fig. S4a). The 3-MA effect on folding capacity may be related with its direct effect on phosphatidylinositol 3-phosphate (PtdIns3P) formation, shown to be required for the

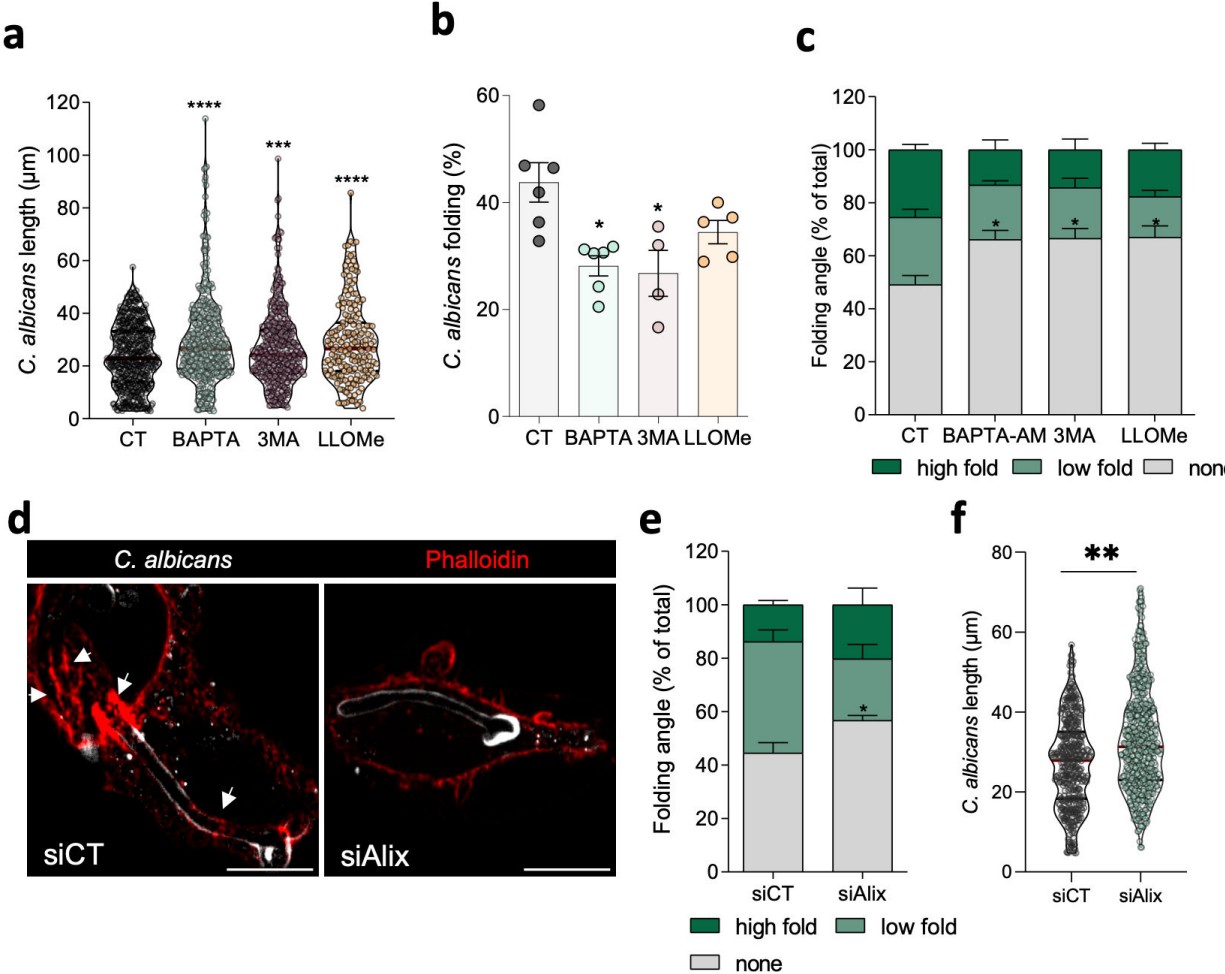

**FIG 3** Integrity of phagosomal membrane impacts *C. albicans* hyphal folding. *C. albicans* length (a) and percentage of hyphal folding (b) upon RAW cells infected with *C. albicans* and treated with vehicle, BAPTA-AM, 3-MA, or LLOMe during 180 min. Results represent the mean ± SEM from at least four independent experiments, respectively. At least 104 cells were analyzed per condition in each independent experiment. *P*-values were calculated by one-way analysis of variance (ANOVA) followed by Dunnett's multiple comparisons test (****$P < 0.0001$; ***$P < 0.001$), compared to control (CT). (c and d) Quantification of hyphal folding angle, upon RAW cells infected with *C. albicans* during 180 min and treated with vehicle, BAPTA-AM, 3-MA, or LLOMe (c) and in condition of Alix depletion (siAlix) (d). Results represent the mean ± SEM from three independent experiments. *P*-values were calculated by a one-way ANOVA followed by Tukey multiple comparisons test (* $P < 0.05$), compared to CT. (e) *C. albicans* length in siAlix cells infected with *C. albicans*. Results represent the mean ± SEM from three independent experiments. *P*-values were calculated by a *t*-test (** $P < 0.01$) compared to CT. (f) Representative image of RAW cells with (siAlix) or without (siCT) Alix depletion infected with *C. albicans*. Arrows point to actin ring structures around the phagosome. Scale bar, 10 µm.

formation of actin puncta. To further confirm that phagosomal integrity is crucial for hyphal folding, Alix was downregulated to hamper lysosomal repair capacity (Fig. S4b). Accordingly, in cells with lower Alix expression levels, we observed not only a significant reduction of the total hyphal folding capacity (Fig. 3d and e) but also an increase of hyphal length (Fig. 3f), when compared to control cells. Accordingly, silencing of Tsg101 in macrophages also induced a significant reduction of high and low folding, with an increase in the no detectable folding percentage (Fig. S4c and d). Altogether these results suggest that phagosomal membrane integrity is required to allow the folding of intralysosomal hyphae, which is vital for host response against *C. albicans*.

## Cx43 as a new molecular player during *C. albicans* infection in macrophages

Next, we sought to find new molecular players involved in the progressive response of macrophages to the yeast-to-hypha transition, with the focus on the overlap between phagolysosomal and folding-related mechanisms. We analyzed a publicly available data set (GSE111731) that simultaneously assessed host and fungal pathogen temporal transcriptional profiles during macrophage and *C. albicans* interactions (9). Initially, we compared the differences between consecutive time points: 0–60, 60–120, and 120–240 min, identifying 695, 657, and 235 differentially expressed genes (DEGs) (*P*-value < 0.05), respectively (Fig. 4a). The results of the DEGs at each time point are summarized in three volcano plots (Fig. 4b). In parallel with our progressive analysis comparing the different time points, we intersected the common DEGs of macrophages infected with *C. albicans* during different infection times. We found 21 DEGs commonly affected at the three time points (Fig. 4c and d). Next, we used STRING database to integrate known (21 DEGs) and 10 predicted associations for all the identified common genes, including physical and functional interactions (Fig. 4d). The 21 common affected genes were clustered into four groups: cell migration, regulated exocytosis, translational initiation, and I-kappaB kinase/NF-kappaB signaling. As mentioned earlier, hyphae folding is a process that depends on phagosome vesicle integrity and the cytoskeleton network-mediating mechanical forces responsible for the migratory capacity of macrophages (31). Thus, considering this information and the four continuously affected pathways, we intersected all the DEGs from cell migration, such as focal adhesion and cell leading edge, and vesicular structures, including endocytic vesicle and lysosomal membrane gene ontology (GO) cellular component (CC) terms, to further disclose new molecular players that could potentiate macrophage capacity to fold hypha (Fig. S5a). The results revealed that 47 DEGs are shared between vesicular trafficking and cell migration-related pathways. We then observed the STRING network of these genes, which were clustered in four functional groups: locomotion, focal adhesion, actin filament-based process, and endocytosis (Fig. 4f). As depicted in the heatmap from Fig. 4g, the expression profile of the identified 47 genes was found to change during *C. albicans* infection. Interestingly, we identified the GJA1 gene, which encodes Cx43, as one of the putative hits affected during the infection process, with a decrease in the expression levels over time (Fig. 4g). In the functional analysis of our network, Cx43 was associated with locomotion and actin filament-based processes, interacting with endocytic components, such as EHD1, and focal adhesion proteins as integrins. Although Cx43 has been initially described as gap junction protein, studies from our group and others have ascribed to Cx43 additional biological roles, igniting a new paradigm in the field of noncanonical functions of Cx43 in cellular homeostasis (34–36). In addition, increased Cx43 expression has already been associated with the cellular response to an inflammatory stimulus (37–39) being implicated in ATP release (40). Therefore, we proceeded to further explore the putative role of Cx43 in phagosome membrane remodeling-associated macrophage response to expanding hyphae.

## Cx43 is recruited to the hyphae-containing phagolysosomes

Given the importance of actin filaments in hyphal folding (31) and the recent reports associating Cx43 to changes in actin cytoskeleton architecture and intracellular

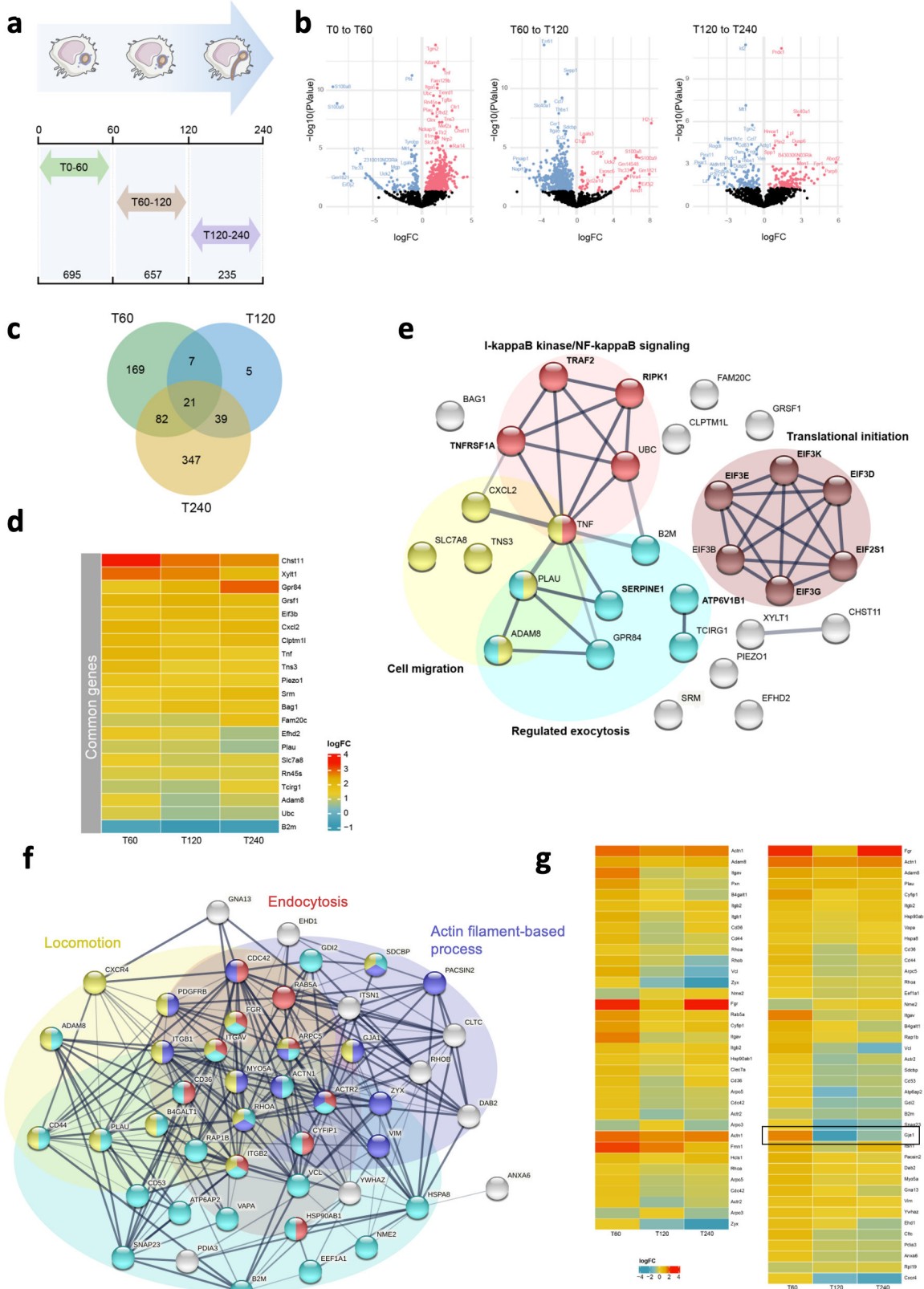

**FIG 4** Cx43 appears as a new molecular player during *C. albicans* infection in macrophages. (a) Schematic representation of our analysis. In order to correlate the different morphological stages of *C. albicans*, a progressive analysis was performed, meaning that each macrophage infection time (T) was compared with the previous one. Through the comparison of T0–60, T60–120, and T120–240 min, 695, 657, and 235 DEGs were identified (*P*-value < 0.05), respectively. (b) Volcano

**FIG 4** (Continued)

plots of differential gene expression during the three-time comparison: T0–60 (left), T60–120 (middle), and T120–40 min (right), with significance [−log10 false discovery rate (FDR] plotted against the log2 relative expression. Black, blue, and red points indicate that genes with no significant difference, significantly increased (FDR < 0.1 and log2 expression ratio >0), and decreased (FDR < 0.1 and log2 expression ratio <0) in expression between the infection time transition (with FDR > 0.05), respectively. (c) Venn diagram illustrates the intersection of DEGs commonly affected in macrophages infected at different times. (d) Heatmap shows the expression levels of the 21 DEGs commonly affected at all the three infection times: T60, T120, and T240 min. Color scheme of gene expression level [log2FC(TPM + 1)] from −1 (blue) to 4 (red). (e) Network analysis and functional enrichment using STRING protein-protein interaction network performed on the 21 DEGs from (c), with an increment of 10 functional interactors. The predicted functional partners are in bold. Significantly modulated pathways and cellular components associated with this network are shown by the different colors (protein-protein interaction enrichment $P$-value, <0.01; strength, >0.7). Rn45s gene was not found in STRING database. (f) Network analysis and functional enrichment of the 47 genes present in the intersection from CC: terms related with vesicular trafficking and cell mobility using STRING database. The functional enrichment of the four main identified clusters is colored in cyan (focal adhesion), red (endocytosis), blue (actin filament-based process), and yellow (locomotion). The solid and the faded lines indicate connection within the same and different cluster, respectively (protein-protein interaction enrichment $P$-value, >0.9). (g) Heatmap representing the relative expression of the DEGs presented in (f).

trafficking (41–43), we hypothesized that Cx43 plays a role during *C. albicans* infection. To address this issue, we started by evaluating the recruitment of Cx43 to *C. albicans* containing phagosomes by confocal microscopy. We observed that upon yeast-to-hypha transition, there was a gradual increase in Cx43 present in vesicular structures enclosing *C. albicans* (Fig. 5a), and this recruitment positively correlates with the length of the internalized *C. albicans,* with an $r^2$ of 0.6456 (Fig. 5b). Additionally, the recruitment of Cx43 to the phagolysosome was also verified when the membrane damage was exacerbated by cotreatment with LLOMe (Fig. S6a). Of note, the Cx43 protein levels were significantly reduced upon 30 min of *C. albicans* infection, followed by a recovery to control levels after 180 min of infection suggesting an increased degradation of Cx43 in the early steps of infection (Fig. S6a and b). These results hint toward a potential role of Cx43 in the macrophage response during the expanding process of the phagosome membrane containing hypha.

## Cx43 promotes hyphal folding

To further elucidate the role of Cx43 during *C. albicans* infection, we generated monoclonal RAW cell lines overexpressing Cx43 (Fig. S7a and b). By using this stable cell line (RAW^Cx43+), we sought to assess the impact of Cx43 expression in hyphal length and folding. Remarkably, a significant decrease in length (Fig. 6a) and an increase in macrophage capacity to fold hyphae (Fig. 6b and c) were observed in cells with higher

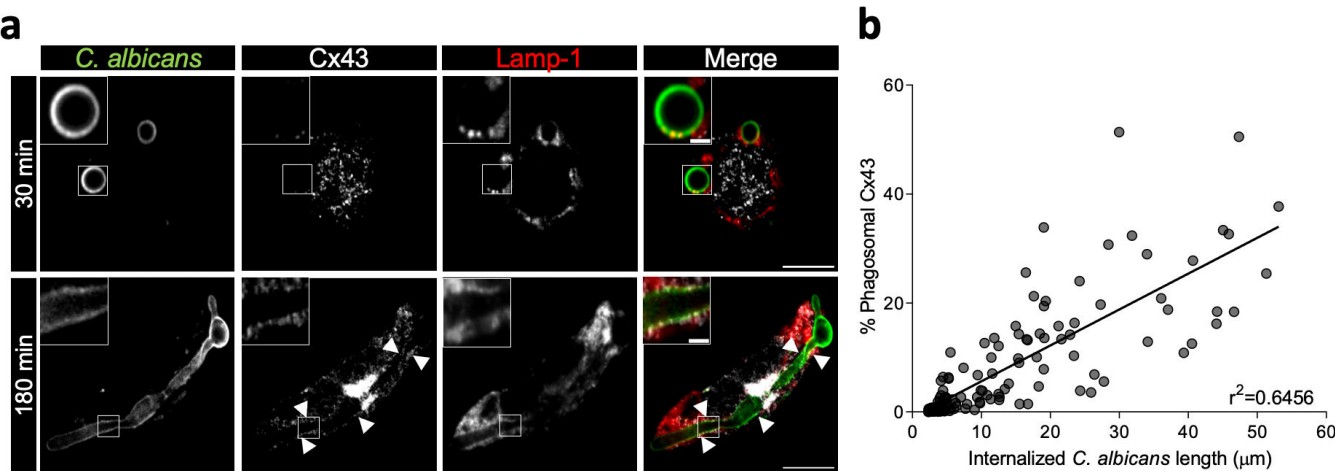

**FIG 5** Cx43 is recruited to the *C. albicans*-containing phagolysosome. (a) Representative confocal images of RAW cells infected with *C. albicans*, fixed after 30 min and 180 min post-infection, and immunostained for *C. albicans*, Cx43, and Lamp-1. Scale bar, 10 µm and 2 µm in the insets. White arrowheads highlight sites of Cx43 accumulation. (b) Regression analysis comparing the length of internalized *C. albicans* and the percentage of phagosomal Cx43 quantified as (phagosomal Cx43/total Cx43)*100 per each infected cell. Results represent the mean ± SEM from three independent experiences.

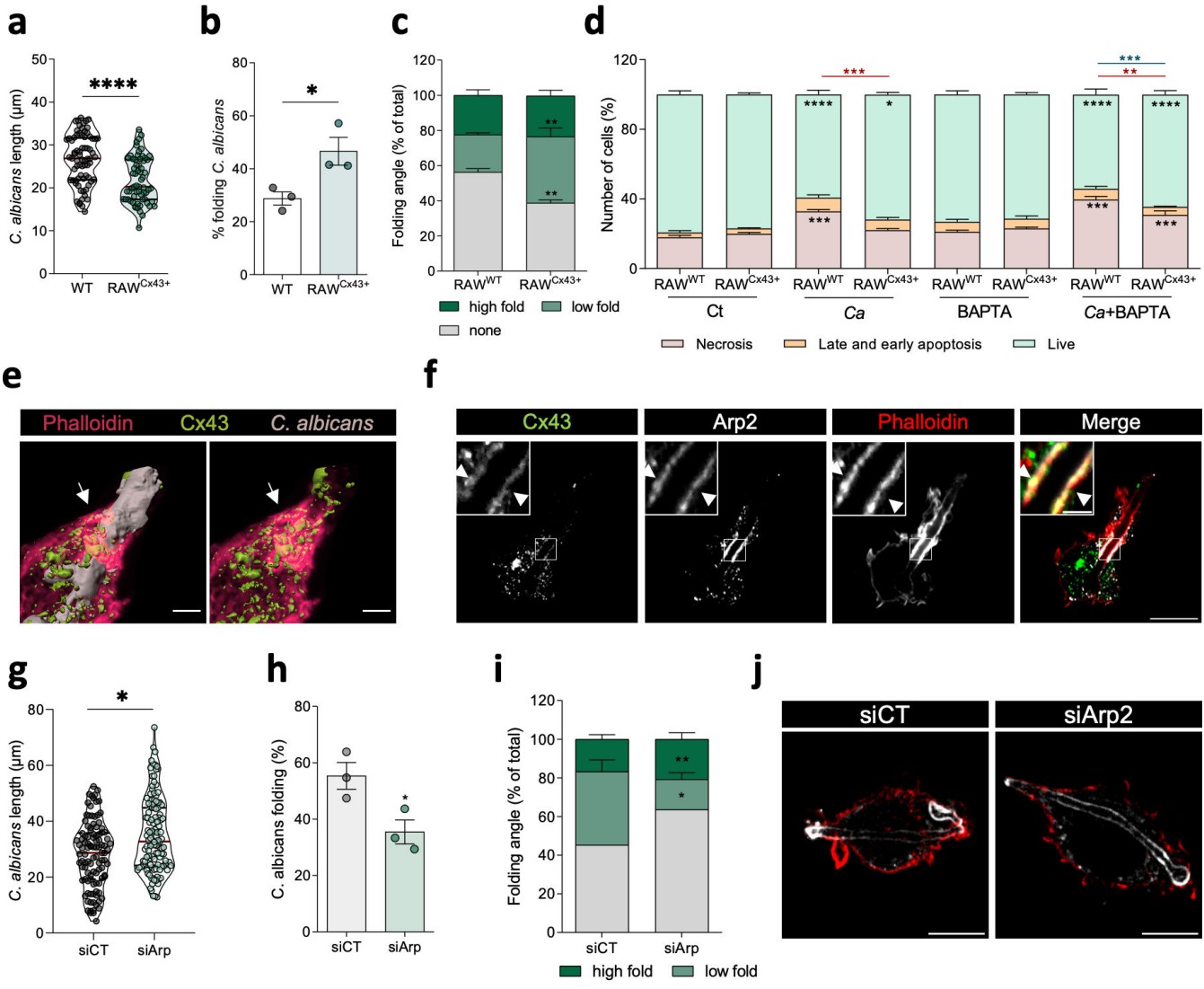

**FIG 6** Cx43 potentiates hyphae folding. (a through c) Quantification of (a) *C. albicans* length, (b) percentage, and (c) angle of hyphal folding upon control RAW cells (RAW^Wt) and cells overexpressing Cx43 (RAW^Cx43+) infected with *C. albicans*. Results represent the mean ± SEM from three independent experiments. (d) Quantification of cell viability of RAW^Wt and RAW^Cx43+ in basal condition, treated with BAPTA-AM or infected with *C. albicans* (*Ca*) during 180 min treated in the presence or absence of BAPTA-AM. Data are means ± SEM, n = 3. Tukey's multiple comparisons test was performed for statistical analysis. Asterisks inside the bars represent the live/death comparison with RAW cells in control condition; the top red and blue asterisks are the comparison between necrosis and live cells, respectively, of RAW^Wt and RAW^Cx43+ cells. (e) Representative three-dimensional image of a RAW cell infected with *C. albicans* for 180 min and immunostained for Cx43 (green), *C. albicans* (beige), and phalloidin (red), reconstructed using Imaris software. Scale bar, 3 µm. (e and f) Representative confocal images of RAW infected with *C. albicans*, fixed after 180 min post-infection, and immunostained for Cx43, Arp2, and phalloidin. Scale bar, 10 µm and 2 µm in the inset images. White arrowheads indicate sites of Cx43 accumulation and colocalization with Arp2. (g) *C. albicans* length in control and RAW cells silenced to Arp2 (siArp2). (h and i) Quantification of hyphal folding (h) percentage and (i) angle, upon infection of control and Arp2-silenced RAW cells with *C. albicans,* during 180 min. *P*-values were calculated by a *t*-test (h) or one-way ANOVA followed by Dunnett's multiple comparisons test (*$P < 0.05$; **$P < 0.01$; ****$P < 0.0001$), compared to control (CT) (i). Results represent the mean ± SEM from three independent experiments. (j) Representative image of siArp2 cells infected with *C. albicans*. Scale bar, 10 µm.

Cx43 levels. In contrast, in conditions where Cx43 levels were diminished, we observed a decrease in macrophage hyphal folding capacity (Fig. S7c and d). To understand the role of Cx43 in this defense mechanism, we next investigated whether Cx43 could counteract the effects of compromised phagosomal membrane integrity during hyphal growth and folding. To induce loss of phagolysosomal membrane integrity, we treated control

and RAW$^{Cx43+}$ with BAPTA-AM to inhibit the repair activity. Strikingly, the presence of increased levels of Cx43 not only reduced *C. albicans* hyphal length but also resulted in a higher percentage of hypha folding in cells treated with BAPTA-AM (Fig. S7e and f). Accordingly, our data demonstrate that, in RAW$^{Cx43+}$ cells, there is a significant increase in the total number of live cells with a concomitant reduction of death cells, when compared with the control cells (Fig. 6d). As expected, when infected cells were exposed to BAPTA-AM, this inhibition of the repair capacity increased cell death percentage. Interestingly, increase in Cx43 levels also counteracted the effects of loss of membrane integrity by increasing the percentage of live macrophages, reinforcing its protective role on cell viability. In RAW$^{Cx43+}$ cells, we also observed that after 180 min, the total amount of extracellular pathogens was reduced when compared with the control (Fig. S7g). These data corroborate previous studies showing that hyphal folding is part of the strategy to constrain this pathogen inside the host cell with consequent reduction of their escape.

As mentioned earlier, nucleation of actin filaments is pivotal to promoting hyphal folding (31). Furthermore, it was previously reported that Cx43 (44) and its GJA1-20k isoform (42) modulate actin polymerization and stabilize filamentous actin. To decipher in more detail the mechanism whereby Cx43 promotes hyphal folding, we next investigated the presence of Cx43 in the actin rings surrounding the hypha. Notably, our results showed a high colocalization of Cx43 with these actin structures (Fig. 6e; Fig. S7h; Movie S1). Considering the network analysis, where Arp2/3 protein complexes were identified as putative hits in the interplay between phagosomal homeostasis and actin-related processes, we determined whether Cx43 found in the actin rings could be implicated in the recruitment of this actin nucleator complex. In agreement, results in Fig. 6f show a robust colocalization of Cx43 and Arp2 around the actin ring structures, which are localized around the phagolysosomes containing *C. albicans*. Moreover, upon depletion of Arp2 using siRNA, we observed a significant increase in hyphae length and decrease in hyphal folding capacity in RAW$^{Cx43+}$ cells when compared to control cells (Fig. 6e through g). Quantification of folding angle revealed a significant reduction of hyphae with low fold in RAW$^{Cx43+}$ cells. To demonstrate that this mechanism is not cell specific, we resorted to a HEK293A cell line overexpressing GFP-Cx43. The results obtained with these cells showed the recruitment of Cx43 to actin structures surrounding frustrated phagosomes (Fig. S7i), previously described in epithelial cells upon hypha invasion (11), suggesting that the recruitment of Cx43 to actin ring structures may be conserved among different cell lines. Overall, the data gathered in this study support a model in which Cx43 has an unanticipated and important role during the formation of actin structures required to induce hyphal folding and prevent *C. albicans* escape from macrophages.

## DISCUSSION

Phagocytic cells depend on several vesicular trafficking events to efficiently restrain, damage, and eliminate pathogens (45). To circumvent the potential deleterious consequences of a defective phagocytic process upon infection, such as that of *C. albicans*, host cells have developed strategies to preserve phagosomal membrane integrity and confine the microorganism in a functional phagolysosomal vesicle. Here, we provide evidence that active phagosomal membrane repair machinery is required to not only support the continuously expanding hypha but also to promote hyphae folding, which was recently described as a new mechanism of host defense. Strikingly, our study also ascribes to Cx43 a novel and unexpected role during host-pathogen response by anchoring actin-nucleator machinery and promoting the fold of the expanding hyphae, protecting host cell viability (Fig. 7).

Emerging studies have shown that the initial Ca$^{2+}$-dependent response after lysosomal damage is important for ESCRT machinery recruitment and consequent repair of minor membrane perturbations induced by pathogens or other lysosomotropic agents [as reviewed in references (46, 47)]. Indeed, the recruitment of the ESCRT-III complex was described in cells exposed to chemical and bacterial-induced lysosomal damage (16, 23,

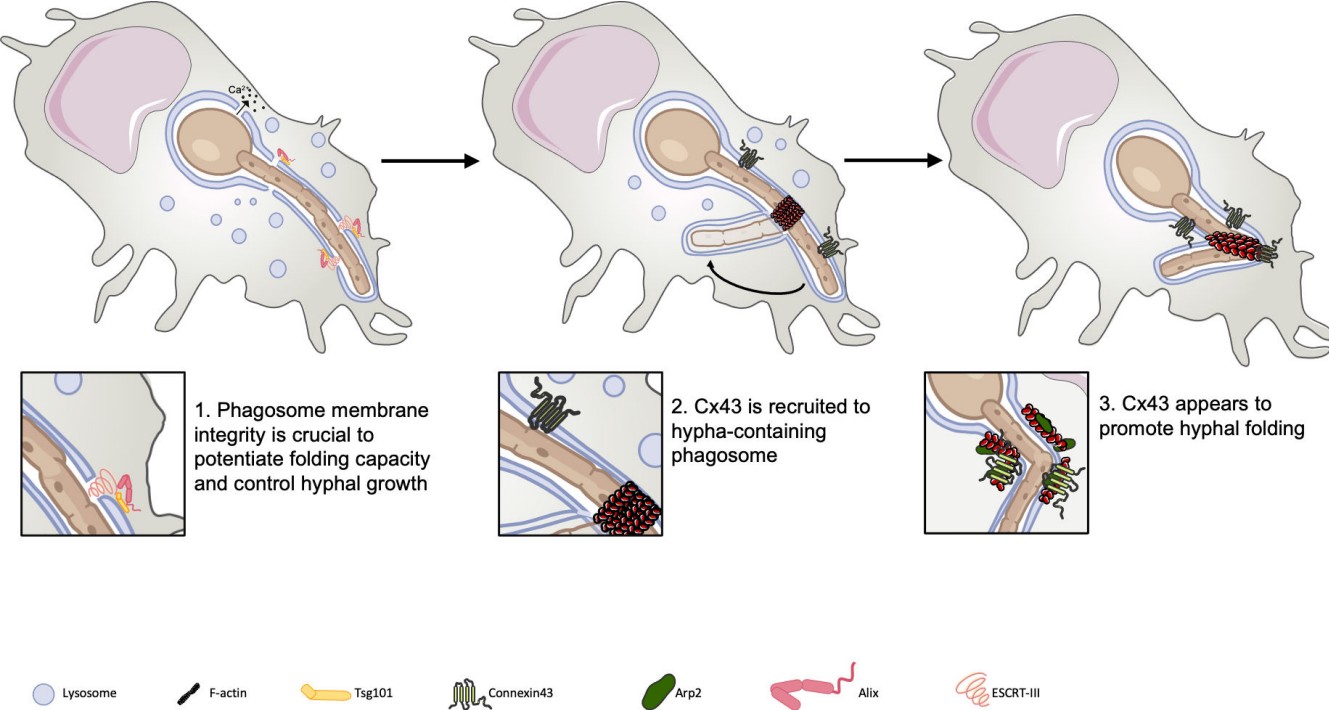

**FIG 7** Schematic summary of Cx43 as a new player in *C. albicans* infection of macrophages. Maintenance of the phagosomal membrane integrity is essential to control the hyphal growth and potentiate the folding capacity of the macrophages, promoting an additional defense and avoiding *C. albicans* escape from the ruptured phagosome. In addition, our work showed that Cx43 is recruited to the phagosome containing hypha and its protein levels seem to affect the hypha folding capacity of macrophages. Interestingly, Cx43 and Arp2 are present in the F-actin contractile rings around the phagosomes containing the hypha and seem to be recruited to the folding sites, suggesting a new role for Cx43 in the *C. albicans* folding process.

48). However, during extreme stretch conditions of phagolysosomes, such as the expanding hyphae-containing phagosomes, the role of the ESCRT components was not clearly elucidated. The data obtained in the present study demonstrate a direct correlation between the recruitment of two components of the ESCRT-III machinery, Alix and Tsg101, and hyphal size, suggesting that an active repair response at the phagosomal membrane is needed to sustain pathogen-containing phagosome growth through a $Ca^{2+}$-dependent process. In accordance with our findings, a recent work demonstrated that the recruitment of repair machinery to the invagination pocket during epithelial infection with *C. albicans* relies on $Ca^{2+}$ levels. Furthermore, this study found that candidalysin secreted by *C. albicans* hyphae induced a cytosolic $Ca^{2+}$ increase, which promoted the hydrolysis of PtdIns(4,5)P2 and loss of cortical actin (26). Notably, a key finding of our study is that reduced lysosomal repair capacity diminished the ability of macrophages to fold the expanding hyphae. Wales et al. previously described a mechanism of cortical actin disassembly induced by acute intracellular $Ca^{2+}$ increase termed as $Ca^{2+}$-mediated actin reset (CaAR) (49). This process leads to transient immobilization of organelles, repair of cortical damage, and the formation of actin-rich protrusions during cell spreading, cell migration, and wound healing (50). Thus, given the existing evidence associating cytosolic $Ca^{2+}$ increase with actin reorganization (43, 51), it is plausible that the local increase of $Ca^{2+}$ released from damaged phagolysosomes can indeed impact the local actin polymerization necessary for hyphal folding.

During the hyphal folding process, adhesion molecules, such as Dectin-1 and β2, as well as cytoskeletal network-mediated mechanical forces responsible for cell migration capacity were found to promote the ability of macrophages to fold and damage the expanding *C. albicans* (31). However, little is known about the molecular players with the capacity to promote actin assembly and modulate phagolysosomal membrane architecture. The analysis of the differentially expressed genes throughout the event of

macrophage infection unveils new putative partners involved in the interplay between phagosomal membrane and actin network-related events. Remarkably, Cx43 emerges as a new player in this biological process. Interestingly, several noncanonical roles beyond gap junction function have been ascribed to Cx43, including its involvement in long-distance intercellular communication (52), selective sorting of microRNA to extracellular vesicles (53), mitochondrial homeostasis (54, 55) and localization (41), and nuclear structure (56). Furthermore, previous studies demonstrated that GJA1-20k, which is mainly formed by the C-terminus of Cx43, can inhibit actin depolymerization and stabilize actin filaments (42, 54) and is able to promote mitochondrial fission by engaging actin to the mitochondria (54). Here, we found that Cx43 is recruited to the expanding phagosome and promotes hyphal fold, possibly by anchoring Arp2. These findings are in accordance with our recent work showing that Cx43 is recruited to damaged lysosomes and promotes exocytosis through the remodeling of actin cytoskeleton (44). In this process, we found that Cx43 effect on actin rearrangement is dependent on Arp2 activity (44). Additionally, supporting our hypothesis, components from this actin nucleator protein were also identified in our STRING analysis, namely, actin-related protein 2/3 complex subunit 5 (ARCP5), a component of the Arp2/3 complex, and actin-related protein 2 (ACTR2), an ATP-binding component of the Arp2/3 complex, promoter of branched actin networks. In the work by Bain et al., the folding process was shown to involve vasodilator-stimulated phosphoprotein actin elongation, and by inhibiting the nucleation with actin inhibitors, cytochalasin D and latrunculin, the folding capacity was reduced. In actin cuffs, observed in epithelial cells infected with *C. albicans* (11), actin assembly was shown to be dependent on formin, which mediates linear actin polymerization and independent of the Arp2/3 complex. Although these actin cuffs share some molecular players with the actin rings characterized in the folding apparatus (31), a direct and functional comparison study was never performed. An obvious difference when we consider these two processes is related to the fact that actin cuffs are assembled on the plasma membrane, while the actin rings inducing hyphae fold are formed in intracellular membranes, i.e., the phagolysosomal membrane. Interestingly, during the mitochondrial fission process, the focal assembly of actin induces local, Drp1-dependent fragmentation of the mitochondrial network in a process dependent on Arp2/3 and formin (57). This suggests that the necessary mechanical forces to induce intracellular membranes stretch can differ from the plasma membrane mechanism. Thus, we believe that Cx43 and Arp2 are both involved in the assembly of actin structures necessary to exert mechanical forces for the stretch and fission of intracellular membrane.

In conclusion, this study demonstrates for the first time that phagolysosomal membrane integrity is necessary to potentiate the hyphal folding process. Furthermore, we show that Cx43 plays a pivotal role during this defense mechanism by contributing to anchor complexes involved in the actin nucleation process and promoting cell host survival.

## MATERIALS AND METHODS

### Antibodies and chemicals

The primary antibodies are as follows: anti-*C. albicans* (catalogue #GTX40096, Genetex), anti-LAMP1 (catalogue #1D4B, Developmental Studies Hybridoma Bank), anti-Alix (catalogue #sc-53540 Santa Cruz Biotechnology), anti-Tsg101 (catalogue #AB83 Abcam), anti-Galectin3 (catalogue #sc-32790 Santa Cruz Biotechnology), anti-LC3 (catalogue #PA1-16930 Thermo Fisher Scientific), anti-p62 (catalogue #51145 Cell Signalling), anti-Beclin1 (catalogue #A-00023 Sigma), and anti-Cx43 (catalogue #AB0016–500 SICGEN). Anti-goat secondary antibodies conjugated with Alexa Fluor 488 and horseradish peroxidase (HRP) were purchased from Invitrogen and Life Technologies Jackson, respectively. Anti-mouse secondary antibodies conjugated with Alexa Fluor 568, Alexa

Fluor 647 and HRP were purchased from Invitrogen and BioRad, respectively. Anti-rabbit secondary antibodies conjugated with Alexa Fluor 488, Alexa Fluor 647, and HRP were purchased from Invitrogen and BioRad, respectively. Anti-rat secondary antibodies conjugated with Alexa Fluor 594 and HRP were purchased from Invitrogen. Phalloidin was obtained from Sigma-Aldrich/Merck. LLOMe (L7393) and 3-MA (M9281-500MG) were purchased from Sigma-Aldrich/Merck. BAPTA-AM was obtained from Calbiochem (196419). Annexin V Alexa Fluor 488 Ready Flow Conjugate was purchased from Thermo Fisher Scientific (R37174).

## Strain and growth conditions

*C. albicans* strain SC5314 was obtained from the Clinical Yeast Culture Collection of the University of Coimbra. Yeast cells were grown at 30°C overnight in yeast extract peptone dextrose (YPD) (0.5% yeast extract, 1% peptone, 2% agar, and 2% glucose) agar plates, harvested by centrifugation, and resuspended in phosphate-buffered saline (PBS) (pH 7.4). Yeasts were heat-killed at 95°C during 30 min and used as an inactive pathogen. The fungal cells were counted in a Neubauer chamber and adjusted to the required cell density.

## Cell lines

Experiments were carried out using the RAW 264.7 macrophage cell line, obtained from the European Collection of Cell Cultures (ECACC 91062702). Generation of RAW cells with increased levels of Cx43 (RAW$^{Cx43}$) is described in the section Cell infection with the lentiviral vector. Generation of the GFP-HEK293$^{Cx43}$ was previously described [58]. RAW 264.7 cells were cultured and maintained in Dulbecco's modified eagle medium (DMEM; Life Technologies, Carlsbad, CA, USA), supplemented with 10% non-inactivated fetal bovine serum (FBS; Gibco, Life Technologies), 3.7 g/L sodium bicarbonate (Thermo Fisher Scientific), 1% penicillin/streptomycin (Pen/Strep 100 U/mL:100 µg/mL; Life Technologies) at 37°C, in a humidified atmosphere with 5% $CO_2$. GFP-HEK293$^{Cx43}$ was grown at 37°C and 5% $CO_2$ in DMEM (Life Technologies, Carlsbad, CA, USA), supplemented with 10% heat-inactivated FBS (Gibco, Life Technologies).

## *C. albicans* infection assays

RAW cells ($1.05 \times 10^5$ cells/cm$^2$) were seeded in 6- or 24-well plates and allowed to adhere and stabilize for 24 h at 37°C, in a humidified atmosphere with 5% $CO_2$. Macrophages were then washed with PBS and infected with *C. albicans* (multiplicity of infection, 1:1), for 1 h at 37°C (pulse), in a humidified atmosphere with 5% of $CO_2$. Then, the non-adherent yeast cells were washed, and *C. albicans* were allowed to internalize for 30 min or 180 min (chase). Cells were treated with 3-MA (20 µM) and CBX (10 µM) 60 min and 10 min prior to infection, respectively, and compounds were maintained during the experiment time. BAPTA-AM (20 µM) was added at the chase time. LLOMe (500 µM) was added 10 min and 30 min before the end of the infection period, 30 and 180 min, respectively. LLOMe was used as positive control for damaged lysosomes. Incubation time points were 30 min, characterized for a higher number of *C. albicans* in the yeast form, and 180 min characterized for an increasing number of *C. albicans* in the hyphal form.

## Colony-forming unit

After *C. albicans* infection internalization for 1 h, non-internalized extracellular fungi were removed with two PBS washes, and cells were further incubated with fresh culture medium without antibiotic during 180 min. *C. albicans* viability was then evaluated using a colony-forming unit (CFU) assay. The supernatants, corresponding to expelled/extracellular *C. albicans*, were collected, and the adhered cells were then lysed with 0.5% Triton X-100 in sterile distilled water and scraped (fraction corresponding to intracellular

fungae). Serial dilutions were performed, and each condition was spread on nutrient agar plates, with colony counting after 24 h at 37°C.

## Immunofluorescence and image acquisition

After each timepoint (30 min or 180 min after infection), the glass coverslips were fixed with 4% paraformaldehyde (pH 7.4) in PBS for 15 min at room temperature. Then, the samples were washed, followed by quenching of the aldehyde groups with ammonium chloride (concentration 50 mM) in 1% PBS for 15 min and afterwards blocked with a solution of 0.05% (wt/vol) saponin and 1% (wt/vol) BSA, in PBS (blocking solution), for 30 min prior to incubation with primary antibodies overnight at 4°C in a humidified chamber. The coverslips were then washed with PBS and incubated with the secondary antibody for 1 h at room temperature, in a humid chamber. Nuclei were stained with 4′,6-diamidino-2-phenylindole dihydrochloride (Sigma-Aldrich). After this, coverslips were washed with PBS and mounted with MOWIOL 4-88 Reagent (Calbiochem, San Diego, CA, USA). The images were collected by fluorescence microscopy using a Zeiss Axio HXP IRE 2 microscope (Carl Zeiss AG) or by confocal microscopy using a Zeiss LSM 710 (Carl Zeiss AG), using 63× oil immersion objective. Images were then analyzed and quantified using Image J-Fiji (described in detail in the section Quantification of the percent of phagosomal proteins) and Imaris softwares. Surface rendering was performed in 8- to 15-µm z-stack images composed of 0.43-µm optical slices using Imaris software (Bitplane AG, Switzerland) and carried out at the Imaging facility at iCBR (iLab-FMUC).

## Quantification of the percentage of phagosomal proteins

The percentage of the proteins in the phagosomal membrane was estimated by measuring the intensity of phagosome-associated proteins per the total cellular intensity of the specific protein multiplied by 100 at different time points (30 min and 180 min post-infection) using ImageJ software. For protein detection, a threshold was applied. Each phagosome was manually designed using the boundaries of *C. albicans* and/or Lamp-1 staining.

## Measurement of hyphal folding

Macrophages with phagocytosed *C. albicans* had its hyphal folding measured and categorized as "none" (no detectable bending or folding), "low fold" (creating a curved hypha or an obtuse angle), or "high fold" (generating an acute angle), using ImageJ software.

## Gene set enrichment analysis

The RNA expression profile of GSE111731 data set was used in this study. The DEGs were obtained by the application of a linear model to the voom-transformed data, using the limma package, for the statistical programming language R. Heatmaps were generated with Complex heatmap package for the statistical programming environment R. The DEGs involved in all the pathways related with vesicles and cell migration GO CC terms were intersected, resulting in 47 common genes. The search tool for the retrieval of interacting genes (STRING) database (11.5) was applied to integrate known protein-protein interactions. The correlation between these proteins was evaluated and displayed.

## Cell infection with the lentiviral vector

RAW cells were incubated with the lentiviral vector pLenti6-CMV-V5-Cx43 and 8 µg/mL of polybrene for 20 min at room temperature. Then, cells were centrifuged for 90 min at 800 × $g$ and 32°C, plated, and incubated with 8 µg/mL of blasticidin. In order to isolate single cell clones from the polyclonal pool, limiting dilution cloning in a 96-well plate was applied by targeting 0.5 cells/well in complete growth medium, then transferring

100 µL of this to each. Plates were then incubated at 37℃, 5% $CO_2$ incubator, and the clonal expansion was monitored during the next days. Overexpression of Cx43 on the monoclonal RAW cell lines was confirmed by Western blot and immunofluorescence analysis.

## siRNA-mediated knockdown

siRNA transfections were performed using Lipofectamine 2000 according to the manufacturer's instructions. siRNA target sequences were Alix (Dharmacon, siGENOME SMARTpool M-004233-02-0005), Cx43 (MISSION esiRNA EHU105621, Sigma-Aldrich), and Arp2 (Thermo Fisher Scientific, s94586). A total of 20 nmol/L siRNA was complexed with the transfection reagent, and an overnight incubation at 37℃ was performed. After that, transfection media were replaced with fresh media, and the experiments were performed after 72 h (Alix and Cx43) or 48 h (Arp2). Non-targeting control sequences (Ambion) were used as controls.

## Immunoblotting

At the end of each timepoint (30 min or 180 min, post-infection), RAW cells were detached and centrifuged at 1,200 rpm, for 5 min, at 4℃. Then, the supernatants were removed, and cell pellets were washed with PBS and again centrifuged at 1,200 rpm, for 5 min, at 4℃. Cell pellets were lysed to obtain total protein extracts using 2× Laemmli buffer. The cell lysates were then sonicated, heated for 5 min at 95℃, and loaded on 10–15% SDS polyacrylamide gels, using Precision Plus Protein Dual Color Standards (BioRad) or GRS Protein Marker MultiColour (GRiSP Research Solutions) as protein ladders. After electrophoresis, proteins were electrotransferred for 2 h at 300 mA with gentle agitation into polyvinylidene difluoride (PVDF) membranes, which were previously activated in methanol. To control protein loading, PVDF membranes were stained with Ponceau S. After washing, membranes were blocked with 1% BSA or 5% of milk (wt/vol) in Tris-buffered saline–Tween 20(TBS-T; 20 mM Tris, 150 mM NaCl, 0.2% [vol/vol] Tween 20, pH 7.6), for 30 min at room temperature and then incubated, overnight at 4℃, with the primary antibodies diluted in blocking solution. Then, the membranes were washed three times with TBS-T for 10 min and incubated with the respective HRP-conjugated secondary antibodies diluted in blocking solution, for 1 h at room temperature. Lastly, the proteins were visualized by chemiluminescence with Clarity Western ECL substrate (Bio-Rad) using ImageQuant LAS 500 (GE Healthcare). The images were quantified using Image Lab 6.0.1 software from BioRad.

## Cell viability

After infection, macrophages were stained with annexin V following the manufacturer's recommendation and analyzed by flow cytometry. Cell populations were divided by live cells [cells negative for annexin V and propidium iodide (PI)], early apoptosis (cells positive for annexin V), late apoptosis (cells positive for annexin V and PI), and necrotic cells (cells only positive to PI). These analyses were performed using FlowJo software.

## Statistical analysis

All data were analyzed using GraphPad Prism Software (version 6.01). The legends of the figures describe the exact number of independent experiences that were analyzed in each experiment. Results were presented as the mean ± standard error of the mean (SEM). Statistical significance was determined using the $t$-test, one-way or two-way ANOVA followed by Dunnett's or Tukey's multiple comparison test, respectively, with a $P < 0.05$, considered statistically significant.

## ACKNOWLEDGMENTS

This work was supported by the European Regional Development Fund (ERDF) through the Operational Program for Competitiveness Factors (COMPETE) under the projects: HealthyAging2020 CENTRO-01-0145-FEDER-000012-N2323, CENTRO-01-0145-FEDER-032179, CENTRO-01-0145-FEDER-032414, POCI-01-0145-FEDER-022122, UIDB/04539/2020; by the Portuguese Foundation for Science and Technology (FCT) under the project POCI-01-0145-FEDER-032414. This project received support from the European Union's Horizon 2020 research and innovation programme under grant agreement MIA-Portugal No 857524 and the Comissão de Coordenação da Região Centro-CCDRC through the Centro2020 Programme.

## AUTHOR AFFILIATIONS

[1]Faculty of Medicine, Coimbra Institute for Clinical and Biomedical Research (iCBR), Clinical Academic Centre of Coimbra (CACC), University of Coimbra, Coimbra, Portugal
[2]Faculty of Medicine, Center for Innovative Biomedicine and Biotechnology (CIBB), University of Coimbra, Coimbra, Portugal
[3]Center for Neurosciences and Cell Biology (CNC-UC), University of Coimbra, Coimbra, Portugal

## PRESENT ADDRESS

Monica Abreu, University of Coimbra, Multidisciplinary Institute of Ageing, Coimbra, Portugal
Neuza Domingues, University of Coimbra, Multidisciplinary Institute of Ageing, Coimbra, Portugal

## AUTHOR ORCIDs

Henrique Girao http://orcid.org/0000-0002-5786-8447

## FUNDING

| Funder | Grant(s) | Author(s) |
| --- | --- | --- |
| MEC | Fundação para a Ciência e a Tecnologia (FCT) | POCI-01-0145-FEDER-032414 | Henrique Girao |
| EC | European Regional Development Fund (ERDF) | UIDB/04539/2020 | Henrique Girao |

## AUTHOR CONTRIBUTIONS

Beatriz Cristovao, Formal analysis, Investigation, Methodology, Writing – original draft | Lisa Rodrigues, Investigation, Methodology, Writing – review and editing | Steve Catarino, Methodology, Resources, Writing – review and editing | Monica Abreu, Data curation, Writing – review and editing | Teresa Gonçalves, Formal analysis, Supervision, Writing – review and editing | Neuza Domingues, Conceptualization, Data curation, Formal analysis, Investigation, Supervision, Visualization, Writing – original draft | Henrique Girao, Conceptualization, Formal analysis, Funding acquisition, Project administration, Supervision, Writing – review and editing

## ADDITIONAL FILES

The following material is available online.

### Supplemental Material

**Supplemental Figures (Spectrum01238-23-s0001.pdf).** Fig. S1 to S7
**Supplemental Movie (Spectrum01238-23-s0002.pptx).** Movie S1

## Open Peer Review

**PEER REVIEW HISTORY (review-history.pdf).** An accounting of the reviewer comments and feedback.

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
