## [Reviewer comments · Microbiology Spectrum]

Microbiology Spectrum

Cx43-mediated hyphal folding counteracts phagosome integrity loss during fungal infection

Henrique Girao, Beatriz Cristovao, Lisa Rodrigues, Steve Catarino, Monica Abreu, Teresa Gonçalves, and Neuza Domingues

Corresponding Author(s): Henrique Girao, University of Coimbra, Faculty of Medicine, Coimbra Institute for Clinical and Biomedical Research

Review Timeline:

Submission Date:	March 23, 2023
Editorial Decision:	April 17, 2023
Revision Received:	July 19, 2023
Accepted:	July 27, 2023

Editor: James Konopka

Reviewer(s): Disclosure of reviewer identity is with reference to reviewer comments included in decision letter(s). The following individuals involved in review of your submission have agreed to reveal their identity: Mira Edgerton (Reviewer #3)

Transaction Report:

DOI: <https://doi.org/10.1128/spectrum.01238-23>

April 17, 2023

Prof. Henrique Girao
University of Coimbra, Faculty of Medicine, Coimbra Institute for Clinical and Biomedical Research
iCBR
Polo III - Health Sciences Campus • Azinhaga Santa Comba, Celas | Portugal
Coimbra 3000-548
Portugal

Re: Spectrum01238-23 (**Cx43-mediated hyphal folding counteracts phagosome integrity loss during fungal infection**)

Dear Prof. Henrique Girao:

The reviewers agreed that this is an interesting manuscript on an important topic. However, two reviewers request modifications to provide additional support for the conclusions.

Link Not Available

Sincerely,

James Konopka

Journals Department
Reviewer comments:

Reviewer #1 (Comments for the Author):

The manuscript "Cx43-mediated hyphal folding counteracts phagosome integrity loss during fungal infection" Cristovao and colleagues investigated whether two proteins which are known to participate in the ESCRT machinery, Alix and Tsg101 (ESCRT-I subunit) mediate phagosome integrity during fungal infection. Furthermore, the authors analyzed Cx43 and postulate that this protein as a putative player in phagosome integrity of macrophage during *C.albicans* infection and yeast to hyphae transition and its phagosomal confinement in vitro.

Major points

- The introduction lacks information on what we know about ESCRT complexes and in particular Alix and Tsg101 during pathogen interaction e.g. phagocytosis. From previous reports, we know that Alix and Tsg101 interact during ESCRT machinery process. The authors should mention that Alix has previously been shown to be highly enriched in macrophage phagosomes (11149929) and that Dill et al. report that ESCRT-I that contains the Tsg101 subunit was found to be enriched in macrophage phagosomes (25755298). Introduction or discussion also lack information about Connexin43 (Cx43) and the contarcicity reports of this protein during infection.
- Fig1: The provided data does not completely support the statement that phagosomal yeast-to-hypha transition of *Candida* increase Alix and Tsg101 presence inside the phagosome. Fig a and b only show the presence of the Alix and Tsg101 at two different time points but difference between them is not clear. Only the regression analysis Fig c and d, which show a correlation between the length of the internalized *Candida* and the percent of both proteins on the phagosome, suggest a phagosomal recruitment. The authors need to purify phagosomes and determine protein levels via immunoblotting. Furthermore, it is surprising that Tsg101 is located mostly around *Candida* surface (yeast) at 30min but not at 180min. Any explanation? Does Tsg101 and Alix are located where the fungus applies a pressure on the membrane or randomly distributed? Do Alix and Tsg101 colocalize during *Candida* phagocytosis?
- Line 120-121: The authors state that the presence of BAPTA-AM, which affect phagosomal membrane integrity, decreases the recruitment of both ESCRT components Alix and Tsg101 to the phagosome. Please provide quantification of the presence of Alix and Tsg101 in BAPTA-AM in comparison with untreated cells. Same issue for Alix and Tsg101 in LLOMe treated cells, no quantification provided.
- Fig 3: 3M is used to inhibit and study the mechanism of autophagy (lysosomal self-degradation) and apoptosis under various conditions. 3-MA inhibits autophagy by blocking autophagosome formation via the inhibition of PI3K. Could the author discuss how blocking autophagy with 3M can affect the folding capacity of macrophage since they find a reduction? Unclear what is significant in Fig. 3c. Low, high or both folds? In Fig3d, it seems that there is more high-fold in the siRNA-Alix than in the siCT. This needs to be discussed. Control Fig3c and Fig 3d are really different in the ratio between low and high. Please explain. What is the role of Tsg101 in hyphal folding? The authors should use Tsg101 siRNA to analyze this.
- Fig4: What is the expression of Alix and Tsg101 in this dataset? Could the author explain the rationale why they picked up GJA1 (Cx43), one of the few gene with decreasing expression over the time course of *Candida*/macrophage interactions (Fig 4g)?
- Fig5: The authors show that Cx43 overexpression reduces hyphal length and increases hyphal folding. Does Cx43 deletion reduces the folding capacity of macrophages (siRNA or pharmacological inhibition)? This is a crucial experiment which will support their overexpression findings.
- Fig7f: the authors used HEK293 cells, which are immortalized human embryonic kidney cells to show co-localization of Cx43 and actin in a non-phagocytic context. I already know that Cx43 has a binding domain for actin (33321985).

Minor points:

- Line 47 "we disclosed a new window of opportunities to disclose the mechanisms underlying the hyphal constraining process"
- Line 178: "but also an increase of hyphal length (Fig. 3f). It is 3e and not 3f. Fig 3f is not described in the manuscript.
- Line 258: In agreement, results in Fig6d show robust colocalization of Cx43 and Arp2. I guess the authors talk about fig 6f
- Line 335: a "that" is missing in the manuscript

Reviewer #2 (Comments for the Author):

In this study Cristovao et al. address the question about the macrophage processes involved in folding of phagocytosed *C. albicans* hyphae. Through elegant and well designed experiments the authors show that phagolysosomal membrane integrity is necessary for the hyphal folding process. Moreover, the authors analyze existing *C. albicans*-macrophage PNSseq datasets, to find and confirm that the gap junction protein Cx43 is critical to facilitate the folding by contributing to anchor complexes involved in the actin nucleation process and promoting cell host integrity. Overall, this study provides valuable information about the repair mechanisms involved in macrophage-*C. albicans* interactions.

Reviewer #3 (Comments for the Author):

This manuscript evaluates the role of macrophage phagosome components' contribution to membrane integrity for its ability to constrain *Candida albicans* hyphal elongation. It is known that phagosome fusion with lysosomes permits vacuole expansion as a result of increased phagocyte load such as *Ca* hyphae and that the cortical actin plays a role in phagosome remodeling. A very interesting hypothesis is that phagosomes membranes may be modified to prevent hyphal disruption by inducing hyphal bending rather than penetration of the membrane.

The initial experiments examine the correlation between hyphal length and Calcium-dependent recruitment of two ESCRT components (Alix and Tsg101). These measures by *r* values are suggestive, but not convincing since these were the only proteins evaluated and total levels of ESCRT proteins were not affected by the presence of *Ca*. Similarly two markers of autophagy/lysophagy were not altered by *Ca*.

Next, hyphal length was examined following treatment with compounds inhibiting phagosome membrane expansion (BAPTA) and repair (3MA) and found that hyphal length was increased (consistent with previous reports), and folding slightly decreased. However a weakness is that no assessment of the impact of elongation/bending is done in terms of macrophage viability or release of Ca.

Next, a public data set is examined to identify additional components that might be involved in phagolysosome remodeling response to Ca. CX43 is identified for further testing. Although known as a gap junction protein, CX43 was shown to be recruited to the phagosome concurrent with Ca hyphae elongation. A functional role for this protein was assessed by overexpression in RAW cells, and found that Ca exhibited decreased hyphal length and increased folding in these cells, and that the viability of CX43 cells was increased. However, this protection seems independent from BAPTA. A significant gap in the data is that there are no experiments showing RAW deleted of CX43 (or siRNA silenced). Arp2 is also shown as a potential partner in this complex, however this data seems incomplete. Also it is not clear how necrosis and apoptosis are being measured in RAW cells and whether this is a result of hyphal membrane penetration. The impact of this study would be strengthened by a more direct assessment of the role of hyphal bending in both short and long-term survival of macrophages.

Staff Comments:

Preparing Revision Guidelines

Please return the manuscript within 60 days; if you cannot complete the modification within this time period, please contact me. If you do not wish to modify the manuscript and prefer to submit it to another journal, please notify me of your decision immediately so that the manuscript may be formally withdrawn from consideration by Microbiology Spectrum.

Dear Editor,

Firstly, we are grateful to you for giving us the opportunity to submit a revised version of the manuscript, taking into consideration the valuable criticisms and suggestions of the reviewers. We do appreciate yours and the reviewers' precious time and efforts in reviewing our paper and providing insightful comments, which decisively contributed to improving the manuscript. We have carefully considered all the comments, incorporated most of the suggestions, and tried our best to answer every point. All the changes in the text are highlighted in yellow in the manuscript file and, below, in black, the point-by-point response to the reviewers' issues. We really hope that this revised version of the manuscript will meet the high standards of Spectrum and be in conditions to be published.

Reviewer comments:

Reviewer #1 (Comments for the Author):

The manuscript "Cx43-mediated hyphal folding counteracts phagosome integrity loss during fungal infection" Cristovao and colleagues investigated whether two proteins which are known to participate in the ESCRT machinery, Alix and Tsg101 (ESCRT-I subunit) mediate phagosome integrity during fungal infection. Furthermore, the authors analyzed Cx43 and postulate that this protein as a putative player in phagosome integrity of macrophage during C.albicans infection and yeast to hyphae transition and its phagosomal confinement in vitro.

We thank the reviewer for the positive comment of our work. We would also like to thank the opportunity to clarify some issues. Please, find below the point-by-point responses to the specific comments.

Major points

-The introduction lacks information on what we know about ESCRT complexes and in particular Alix and Tsp101 during pathogen interaction e.g. phagocytosis. From previous reports, we know that Alix and Tsg101 interact during ESCRT machinery process. The authors should mention that Alix has previously been shown to be highly enriched in macrophage phagosomes (11149929) and that Dill et al. report that ESCRT-I that contains the Tsg101 subunit was found to be enriched in macrophage phagosomes (25755298).

We thank the reviewer for the comment and we have added the following sentence to the revised version of the manuscript.

"Interestingly, some of these repair-related complexes such as ESCRT-I, including Tumor susceptibility gene 101 protein (Tsg101)(1), and ALG-2-interacting protein X (Alix) (2) were already shown to be enriched in phagosomes."

Although we understand the pertinence of the topic, we believe that we should not extend it in the introduction considering that these works focus on the phagocytosis process and use beads to characterize the phagosome proteome, but these procedures did not induce loss of phagolysosome membrane integrity. In the introduction, we have tried to spotlight what is known regarding the molecular machinery and cellular response upon phagosome/lysosome membrane damage, highlighting what is known in pathogen-induced loss of integrity.

Introduction or discussion also lack information about Connexin43 (Cx43) and the contarcicty reports of this protein during infection.

We thank the reviewer for underlying this point. Following the suggestion of the reviewer we added the following sentence to Page 6 lines 218-219:

“In addition, increased Cx43 expression has already been associated with the cellular response to an inflammatory stimulus (3–5) being implicated in ATP release (6).”

- Fig1: The provided data does not completely support the statement that phagosomal yeast-to-hypha transition of Candida increase Alix and Tsg101 presence inside the phagosome. Fig a and b only show the presence of the Alix and Tsg101 at two different time points but difference between them in not clear. Only the regression analysis Fig c and d, which show a correlation between the length of the internalized Candida and the percent of both proteins on the phagosome, suggest a phagosomal recruitment. The authors need to purify phagosomes and determine protein levels via immunoblotting.

We thank the reviewer for raising this very important and pertinent issue. We do apologize if we were not clear in transmitting our aim and the results obtained. The two different infection time points (30 and 180 min) used in this study were selected to increase the probability of having higher levels of macrophages with *C. albicans* in yeast form or shorter hyphae, at 30 min, and more elongated hyphae, at 180min. Our findings demonstrate that the total amount of Alix and Tsg101 recruited to the phagosomal membrane (not inside) is increased upon the transition of yeast-to-hypha. Furthermore, we observed a continuous phagosomal recruitment of these proteins that is directly proportional to phagosome length. This means that as the total area of the extending phagosome augments there is also an increased accumulation of repair machinery components. To be clearer, we now rephrase line 114 from page 4 to: “leads to a total increase of Alix and Tsg101 in phagolysosomes containing hyphae”, in the revised version.

We agree that alternative assays to corroborate the obtained findings would improve the robustness of our results. We tried to use a lysosomal enrichment kit to measure Tsg101 and Alix in the protein extracts (Figure 1 from this letter). However, we did not succeed in obtaining a clean lysosomal fraction (maybe related with likely changes to lysosomal density upon infection) and we were not able to observe changes between the conditions (we used LLOMe as a positive control to induce lysosomal damage). Although this can theoretically be an interesting and suitable methodology that we could use to isolate pathogen-containing phagosomes, we think this type of lysosomal

isolation approaches likely do not reflect or respond to our question since: (i) after infection, not all cells are infected with *C. albicans* resulting in a relatively low percentage of cells with pathogen-containing phagosomes; (ii) the majority of cells are infected with only one pathogen giving rise to a single phagosome among several lysosomes; (iii) the correlation between the size of *C. albicans* and the recruitment of the proteins cannot be accurately assessed, considering that at 30min of infection we already observe several hyphae or that after 180min of infection some *C. albicans* are still in yeast form and iv) the presence of *C. albicans* phagolysosomes is likely to change their density which is plausible to interfere with the isolation procedure. Thus, taking all these technical and methodological constraints into consideration, we think that the microscopy approach we used, inspired by the work from Westman et al, published in Cell Reports (7) is the best approach to investigate the cellular mechanisms activated during the elongation of the phagosome upon yeast-to-hypha transition.

Figure 1 – Immunoblot from lysosomal enrichment extracts isolated from control, *C. albicans* infected cells (180min) and macrophages treated with LLOMe for 30min.

Furthermore, it is surprising that Tsg101 is located mostly around Candida surface (yeast) at 30min but not at 180min. Any explanation?

We completely agree with the reviewer's concern, which was regrettably caused by the image we selected for the manuscript. Indeed, the representative image for Tsg101 was not appropriated since there is an obvious autofluorescence or bleeding through signal from *C. albicans* staining into Tsg101. Importantly, the Tsg101 staining is punctuated and not continuous as observed in panel b from Figure 1. We do apologize for the mistake, and we have now replaced the panel with a better image in the revised version of the manuscript.

Does Tsg101 and Alix are located where the fungus applies a pressure on the membrane or randomly distributed?

We acknowledge the reviewer for this very relevant question. Although we are not able to accurately and assertively respond to this question, our data demonstrate that in general there is a random distribution of Alix and Tsg101 around the phagosome. Nevertheless, in some macrophages infected with more elongated hypha an increase in the repair machinery puncta at the tips of the hyphae can be also clearly observed.

Do Alix and Tsg101 colocalize during Candida phagocytosis?

To answer this question, we performed double staining of Tsg101 and Alix (Figure 2 in this letter, below) after which we quantified the colocalization between these two proteins using Fiji software. Our results show a colocalization of Tsg101 and Alix with a Manders coefficient of 0.461 ± 0.09 (mean \pm SEM from 10 cells, in 3 independent experiments).

Figure 2 – Representative image of Tsg101 and Alix colocalization in the phagolysosome. Scale bar, 10 μ m and 2 μ m in the inset images.

- Line 120-121: The authors state that the presence of BAPTA-AM, which affect phagosomal membrane integrity, decreases the recruitment of both ESCRT components Alix and Tsg101 to the phagosome. Please provide quantification of the presence of Alix and Tsg101 in BAPTA-AM in comparison with untreated cells. Same issue for Alix and Tsg101 in LLOMe treated cells, no quantification provided.

Thank you very much for the suggestion and we do apologize for not having included these data in the first submission. We have now added quantifications of the percentage of Alix and Tsg101 in phagosomes when cells were treated with BAPTA-AM and LLOMe to Figure S1, where we observed a statistically significant reduction of Alix and Tsg101 recruitment in the presence of BAPTA, whereas in cells treated with LLOMe the levels of phagosomal Alix and Tsg101 increase.

- Fig 3: 3M is used to inhibit and study the mechanism of autophagy (lysosomal self-degradation) and apoptosis under various conditions. 3-MA inhibits autophagy by blocking autophagosome formation via the inhibition of PI3K. Could the author discuss how blocking autophagy with 3M can affect the folding capacity of macrophage since they find a reduction? Unclear what is significant in Fig. 3c. Low, high or both folds? In Fig3d, it seems that there is more high-fold in the siRNA-Alix than in the siCT. This needs to be discussed. Control Fig3c and Fig 3d are really different in the ratio between low and high. Please explain. What is the role of Tsg101 in hyphal folding? The authors should use Tsg101 siRNA to analyze this.

We thank the reviewer for all the pertinent questions. Despite the number of studies trying to unveil the mechanisms inside the phagosome triggered by *C. albicans* infection, there are still many open questions, such as, for example the host-membrane sources that quickly contribute for phagosomal membrane growth. Additionally, the mechanism whereby phagosomal membrane remodeling can influence the newly described host strategy of actin-mediated hyphal fold that is assembled on the phagosome membrane remains unknown. Thus, one of our main aims was to assess the impact of autophagic machinery upon phagosomal membrane elongation and its folding process. With the strategies we used we did not detect a significant recruitment of autophagic machinery to the pathogen-containing phagolysosome, however we did observe some impact in the folding capacity of the phagosome when autophagy was blocked. It is conceivable that the effect of 3-MA on the folding capacity of macrophages is related to its impact on Phosphatidylinositol 3-phosphate (PtdIns3P) and not on autophagy activity. Indeed, it was demonstrated that PtdIns(3)P is required for the formation of actin puncta, and the inhibition of this phosphoinositide formation by 3-MA treatment, by blocking PI(3)K activity, diminished the formation of actin puncta [ref-<https://www.nature.com/articles/ncb3215>]. Thus, despite being an interesting result that deserves further investigation, our main focus is the link between membrane integrity and folding capacity.

To clarify this issue, we now added the following sentence to the results section:

“The 3-MA effect on folding capacity may be related with its direct effect on phosphatidylinositol 3-phosphate (PtdIns3P) formation, shown to be required for the formation of actin puncta.” Highlighted in yellow.

Unclear what is significant in Fig. 3c.

Thank you for raising this point. We have now included in the figure legend the following sentence:

“In panels c-e statistics were performed against the same type of folding (none, low or high) in the control cells versus the treated-groups.”

Low, high or both folds?

We performed the quantification of the hyphae folding capacity of macrophages based on the previous work from Bain et. al (8): “none (no detectable bending or folding); moderate (creating a curved hypha or an obtuse angle), or high (generating an acute angle).” . The description of this quantification is present in page number 12, lines 444-146.

In Fig3d, it seems that there is more high-fold in the siRNA-Alix than in the siCT. This needs to be discussed.

We acknowledge the reviewer for this valuable comment. Although it seems there is more high-folding in cells silenced for Alix, this variation is not statistically different as can be confirmed in the test analysis shown in Figure 3 in this letter.

Unpaired t test		
Tabular results		
1	Table Analyzed	sialix high
2		
3	Column B	sialix
4	vs.	vs.
5	Column A	sict
6		
7	Unpaired t test	
8	P value	0.2390
9	P value summary	ns
10	Significantly different (P < 0.05)?	No
11	One- or two-tailed P value?	Two-tailed
12	t, df	t=1.383, df=4
13		
14	How big is the difference?	
15	Mean of column A	13.84
16	Mean of column B	21.87
17	Difference between means (B - A) ± SEM	8.026 ± 5.805
18	95% confidence interval	-8.092 to 24.14
19	R squared (eta squared)	0.3233
20		

Figure 3 - Statistic analysis between the percentage of high fold from siCT and siAlix.

Control Fig3c and Fig 3d are really different in the ratio between low and high. Please explain.

We do agree with the reviewer’s comment, and thank you for highlighting it. Although we observed some differences between control conditions in both experiments, these are not statistically different. However, it is important to stress that control cells from panel Fig. 3e were treated with scramble siRNA and lipofectamine that, although is not

expected to considerably affect the cellular response to *C albicans* infection, we cannot exclude that this procedure can interfere with the properties of the lipidic bilayers, and consequently with the folding capacity of macrophages. For this reason, we used complementary pharmacological and genetic approaches to cohesively strengthen our results and support our conclusions.

Please explain. What is the role of Tsg101 in hyphal folding? The authors should use Tsg101 siRNA to analyse this.

The role of Tsg101 recruitment to hyphae-containing phagosomes and the consequent impact on their folding is expected to be similar to the role of Alix or of calcium signalling promoted by membrane damage to these vesicles. Additionally, it is anticipable that inhibition of phagosome-membrane repair capacity during hyphal elongation, through Tsg101 silencing, will compromise the proper assembly of the actin ring surrounding the vesicle, and thus result in a decrease in the folding capacity of macrophages. To further reinforce our hypothesis, in the revised version of this manuscript we have included the quantification of hyphal folding percentage upon Tsg101 depletion to **Fig. S4 c-d** as well as the following sentence (Page 5, line 180-182):

“Accordingly, silencing of Tsg101 in macrophages also induced a significant reduction of high and low folding, with an increase in the no detectable folding percentage (**Fig. S4c-d**).”

- Fig4: What is the expression of Alix and Tsg101 in this dataset? Could the author explain the rational why they picked up GJA1 (Cx43), one of the few gene with decreasing expression over the time course of Candida/macrophage interactions (Fig 4g)?

We acknowledge the reviewer for having raised this valuable and relevant question. Although Alix and Tsg101 were not detected as differentially expressed genes in this dataset, we assessed the total levels of these proteins and no significant differences were observed in the levels of Tsg101 and Alix (Figure 4, only for the reviewers).

Among all the obtained hits, we picked Cx43 and proceeded with studies with this protein taking into consideration the recent and emergent concept of non-canonical roles of Cx43 in cellular homeostasis and trafficking, which ascribes to Cx43 other biological functions beyond gap junction mediated communication. Furthermore, in recent years we have been developing several studies to unravel these unconventional roles of Cx43. In one of our recent works, we have shown that Cx43 is recruited to damaged vesicles and has an impact on lysosomal exocytosis by promoting actin remodelling (9). Thus, having also taken into account that lysosome exocytosis is directly related with increased cellular migratory capacity (9) as well as actin remodelling (9, 10), we thought that Cx43 could also be a promising hit to explore in the *C. albicans* folding mechanism.

Figure 4 – Expression levels of Alix and Tsg101. Representative immunoblot image (top left) and quantification (top right) of total Alix protein levels upon RAW cells infection with *C. albicans* (Ca) or heat-killed (HK) *C. albicans*, or incubated with LLOMe, for 180 min. Results represent the mean \pm SEM from 3 independent experiments. Ponceau was used as a loading control. Representative immunoblot image (bottom left) and quantification (bottom right) of total Tsg101 protein levels upon RAW cells infection with Ca or HK, or incubated with LLOMe, for 180 min post-infection. Results represent the mean \pm SEM from at least 3 independent experiments. Ponceau was used as a loading control.

- Fig5: The authors show that Cx43 overexpression reduces hyphal length and increases hyphal folding. Does Cx43 deletion reduces the folding capacity of macrophages (siRNA or pharmacological inhibition)? This is a crucial experiment which will support their overexpression findings.

We would like to thank the reviewer for this excellent suggestion. We have performed Cx43 silencing in macrophages after which we assessed pathogen-containing phagolysome folding. Our new data show that reduction of Cx43 levels in RAW cells decreases their capacity to induce hyphal folding. We have now included the quantification of the macrophage's capacity to fold hyphae in supplementary Fig. S7c, as well as the following sentence in page 7, lines 243-244.

"In contrast, in conditions where Cx43 levels were diminished, we observed a decrease in macrophage hyphal folding capacity (**Fig. S7c-d**)."

- **Fig7f: the authors used HEK293 cells, which are immortalized human embryonic kidney cells to show co-localization of Cx43 and actin in a non-phagocytic context. Already known that Cx43 has a binding domain for actin (33321985).**

One of the main objectives of this study is to elucidate the role of Cx43 in macrophage response to pathogen infection. We resorted to HEK293 cells to confirm and expand some of the results obtained in macrophages. Although these are cells with no phagocytic activity, they were previously described to form frustrated phagosomes with actin structures similar to the actin rings described in macrophages(11). As we demonstrated in the macrophage cell line, we also observed increased colocalization between Cx43 and actin in the actin cup assembly region.

Minor points:

-Line 47 "we disclosed a new window of opportunities to disclose the mechanisms underlying the hyphal constraining process"

-Line 178: "but also an increase of hyphal length (Fig. 3f). It is 3e and not 3f. Fig 3f is not described in the manuscript.

-Line 258: In agreement, results in Fig6d show robust colocalization of Cx43 and Arp2. I guess the authors talk about fig 6f

-Line 335: a "that" is missing in the manuscript

We thank you for all the minor comments that are only possible due to the thorough revision performed by the reviewer. These suggestions/corrections have been included and contribute to improve the quality of the revised version of this manuscript. We have now corrected the text and also the order of the panels in Fig. 3.

Reviewer #2 (Comments for the Author):

In this study Cristovao et al. address the question about the macrophage processes involved in folding of phagocytosed C. albicans hyphae. Through elegant and well designed experiments the authors show that phagolysosomal membrane integrity

is necessary for the hyphal folding process. Moreover, the authors analyze existing *C. albicans*-macrophage PNSseq datasets, to find and confirm that the gap junction protein Cx43 is critical to facilitate the folding by contributing to anchor complexes involved in the actin nucleation process and promoting cell host integrity. Overall, this study provides valuable information about the repair mechanisms involved in macrophage-*C. albicans* interactions.

We thank the reviewer for the positive and constructive comment of our work and on its conceptual originality.

Reviewer #3 (Comments for the Author):

This manuscript evaluates the role of macrophage phagosome components' contribution to membrane integrity for its ability to constrain *Canada albicans* hyphal elongation. It is known that phagosome fusion with lysosomes permits vacuole expansion as a result of increased phagocyte load such as Ca hyphae and that the cortical actin plays a role in phagosome remodeling. A very interesting hypothesis is that phagosomes membranes may be modified to prevent hyphal disruption by inducing hyphal bending rather than penetration of the membrane.

We thank the Reviewer for the thorough assessment of our study as well as all the suggestions that will contribute to improving the quality of the manuscript. Please find below a point-by-point rebuttal to the specific comments. We hope the reviewer finds our clarifications satisfactory.

The initial experiments examine the correlation between hyphal length and Calcium-dependent recruitment of two ESCRT components (Alex and Tsg101). These measures by *r* values are suggestive, but not convincing since these were the only proteins evaluated and total levels of ESCRT proteins were not affected by the presence of Ca. Similarly two markers of autophagy/lysophagy were not altered by Ca.

We acknowledge the valuable and very pertinent comment from the reviewer. During cellular trafficking events there are several kiss-and-run fusion and fission processes between vesicular membranes as well as the recruitment of different effectors that sustain vesicular remodelling and maturation. ESCRT machinery, first described in MVB biogenesis, is one example of a protein complex that is implicated in these events, being one of the first proteins to be recruited to ensure membrane dynamics, integrity, shape and organelle/ structure function. Thus, it is not surprising that no detectable changes in the total levels of these trafficking proteins are found, but instead a redistribution of their cellular localization. The same cellular response can be considered for autophagic proteins. Moreover, in our study we only quantified two proteins from ESCRT and autophagic machinery because the methodology associated with these quantifications is technically challenging and difficult. Nevertheless, we agree that more proteins and/ or other alternative assays to corroborate these findings would improve the robustness of our results. Other alternatives to quantify the

recruitment of these proteins would be using phagosome enriched fractions or lysolP. We tried to use this type of approach by measuring Tsg101 and Alix in the protein extracts of a lysosomal enriched pool fraction (Figure 1 from this letter). However, we were not successful in obtaining a clean lysosomal fraction (perhaps due to changes in lysosomal density upon Ca infection) or in observing changes between the conditions (we used LLOMe as a positive control for inducing lysosomal damage). Although this can theoretically be an interesting and suitable methodology which we could use to isolate pathogen-containing phagosomes, we think this type of lysosomal isolation approach would likely not reflect or provide answers to our question since: (i) after infection, not all cells are infected with *C. albicans* resulting in a relatively low percentage of cells with pathogen-containing phagosomes; (ii) the majority of cells are infected with only one pathogen giving rise to a single phagosome among several lysosomes; (iii) the correlation between the size of *C. albicans* and the recruitment of the proteins cannot be accurately assessed, considering that at 30min of infection we already observe several hyphae or that after 180min of infection some *C. albicans* are still in yeast form and iv) the presence of *C. albicans* in phagolysosomes is likely to change their density which is plausible to interfere with the isolation procedure. Thus, taking into consideration all of these technical and methodological constraints, we think that the microscopy approach we used, inspired by the work from Westman et al, published in Cell Reports (7) is the best approach to investigate the cellular mechanisms activated during the elongation of the phagosome upon yeast-to-hypha transition (7).

Next, hyphal length was examined following treatment with compounds inhibiting phagosome membrane expansion (BAPTA) and repair (3MA) and found that hyphal length was increased (consistent with previous reports), and folding slightly decreased. However a weakness is that no assessment of the impact of elongation/bending is done in terms of macrophage viability or release of Ca.

We thank the reviewer for this excellent suggestion. We have tried to correlate folding with cell viability using annexin and PI staining, however it was not possible to combine the fluorescence channels used to label actin and *C. albicans* to directly correlate host viability with hyphal folding. However, our data in Fig. 6d already shows that BAPTA, which increases *C. albicans* length and decreases their folding, promotes increased levels of cellular necrosis after infection. Moreover, in the revised version of the manuscript we included new data regarding the escape of *C. albicans* using the colony forming units (CFUs) methodology. We performed this in the wildtype RAW cell line and RAW with increased levels of Cx43. Strikingly, this new data reveals that increased levels of Cx43, which promotes hyphal fold, decreases the capacity of *C. albicans* to escape, reducing the number of CFUs resulting from the extracellular fraction (Fig S7g). Accordingly, we added the following sentence to the revised version of the manuscript:

“In RAW^{Cx43+} cells, we also observed that after 180min the total amount of extracellular pathogens is reduced when compared with the control (**Fig. S7g**). These data corroborate previous studies showing that hyphal folding is part of the strategy to constrain this pathogen inside the host cell with a consequent reduction of their escape.”

Next, a public data set is examined to identify additional components that might be involved in phagolysosome remodeling response to Ca. CX43 is identified for further testing. Although known as a gap junction protein, CX43 was shown to be recruited to the phagosome concurrent with Ca hyphae elongation. A functional role for this protein was assessed by overexpression in RAW cells, and found that Ca exhibited decreased hyphal length and increased folding in these cells, and that the viability of CX43 cells was increased. However, this protection seems independent from BAPTA. A significant gap in the data is that there are no experiments showing RAW cells deleted of CX43 (or siRNA silenced).

We would like to thank the reviewer for this excellent suggestion. To address this issue we have performed new experiments in which we assessed pathogen-containing phagolysosome folding in macrophages depleted of Cx43. Our new data show that reduction of Cx43 levels in RAW cells decrease their capacity to induce hyphal folding (Fig. S7c-d). We have now included this quantification of the macrophage's capacity to fold hyphae in supplementary Fig. S7d, as well as the following sentence on page 7, lines 243-244.

"In contrast, in conditions where Cx43 levels were diminished, we observed a decrease in macrophage hyphal folding capacity (**Fig. S7c-d**)."

Arp2 is also shown as a potential partner in this complex, however this data seems incomplete.

We acknowledge the reviewer for this very relevant comment. However, we did not explore the role of Arp2 as a new interaction partner of Cx43 in more detail because this mechanism is thoroughly described in our more recent work available as a preprint version that is now under revision (12). Here, we report that Arp2 interacts with Cx43 and that this interaction is responsible for the observed effect of Cx43 upon cytoskeleton remodeling and lysosomal exocytosis.

Also it is not clear how necrosis and apoptosis are being measured in RAW cells and whether this is a result of hyphal membrane penetration. The impact of this study would be strengthened by a more direct assessment of the role of hyphal bending in both short and long-term survival of macrophages.

We thank the reviewer for this excellent comment and suggestion. Necrosis was measured using the cell viability kit Annexin V Alexa Fluor™ 488 Ready Flow Conjugate. We have now added the following to the methods section:

"Cell viability

After infection, macrophages were stained with annexin V following the manufacturer's recommendations and analysed by flow cytometry. Cell populations were divided by live cells (cells negative for annexin V and propidium iodide, PI), early apoptosis (cells positive for annexin V), late apoptosis (cells positive for annexin V and PI) and necrotic cells (cells only positive for PI). This analysis was performed using FlowJo software."

We do agree that establishing a direct correlation between the folding capacity of host cells and their survival would be an important point to address. However, as aforementioned, we do not have any approach that allows us to infer about this direct correlation. Measuring and comparing the survival percentage of infected RAW cells with higher or lower levels of Cx43 or treated either in the presence or absence of BAPTA only allowed us to have an indirect measurement of this correlation. Nevertheless, we have now assessed the effect of Cx43 on the escape of *C. albicans* through the measurement of CFUs (Fig. S7g in the revised version). We added the following sentence to page 7, lines 255-259:

“In RAW^{Cx43+} cells, we also observed that after 180min the total amount of extracellular pathogens is reduced when compared with the control (**Fig. S7g**). These data corroborate previous studies showing that hyphal folding is part of the strategy to constrain this pathogen inside the host cell, with consequent reduction of their escape.”

References

1. Dill BD, Gierlinski M, Hä Rtlöva A, Gonzá Lez Arandilla A, Guo M, Clarke RG, Trost M. 2015. Quantitative Proteome Analysis of Temporally Resolved Phagosomes Following Uptake Via Key Phagocytic Receptors* □ S. *Molecular & Cellular Proteomics* 14:1334–1349.
2. Garin J, Diez R, Kieffer S, Dermine JF, Duclos S, Gagnon E, Sadoul R, Rondeau C, Desjardins M. 2001. The phagosome proteome: insight into phagosome functions. *J Cell Biol* 152:165–180.
3. Jara PI, Boric MP, Sa'ez JC. 1995. after activation with lipopolysaccharide and appear to form gap junctions with endothelial cells after ischemia-reperfusion (intercellular communication/inflammation/polymorphonuclear granulocytes). *Cell Biology Leukocytes express connexin* 92:43.
4. Eugenín EA, Brañes MC, Berman JW, Sáez JC. 2003. TNF- α Plus IFN- γ Induce Connexin43 Expression and Formation of Gap Junctions Between Human Monocytes/Macrophages That Enhance Physiological Responses. *The Journal of Immunology* 170:1320–1328.
5. Li W, Bao G, Chen W, Qiang X, Zhu S, Wang S, He M, Ma G, Ochani M, Al-Abed Y, Yang H, Tracey KJ, Wang P, D'angelo J, Wang H. 2018. Connexin 43 Hemichannel as a Novel Mediator of Sterile and Infectious Inflammatory Diseases Cytoplasmic membrane-bound connexin 43 (Cx43) proteins oligomerize into hexameric channels (hemichannels) that can sometimes dock with hemichannels on adjacent cells to form gap junctional (GJ) channels. *SciENTific RepoRTs* | 8:166.
6. Dosch M, Zindel J, Jebbawi F, Melin N, Sanchez-Taltavull D, Stroka D, Candinas D, Beldi G. 2019. Connexin-43-dependent ATP release mediates macrophage activation during sepsis. *Elife* 8.

7. Westman J, Walpole GFW, Kasper L, Xue BY, Elshafee O, Hube B, Grinstein S. 2020. Lysosome Fusion Maintains Phagosome Integrity during Fungal Infection. *Cell Host Microbe* 1–15.
8. Bain JM, Alonso MF, Childers DS, Walls CA, Mackenzie K, Pradhan A, Lewis LE, Louw J, Avelar GM, Larcombe DE, Netea MG, Gow NAR, Brown GD, Erwig LP, Brown AJP. Immune cells fold and damage fungal hyphae <https://doi.org/10.1073/pnas.2020484118/-/DCSupplemental>.
9. Kay RR, Langridge P, Traynor D, Hoeller O. 2008. Changing directions in the study of chemotaxis. *Nature Reviews Molecular Cell Biology* 2008 9:6 9:455–463.
10. Bergert M, Chandradoss SD, Desai RA, Paluch E. Cell mechanics control rapid transitions between blebs and lamellipodia during migration <https://doi.org/10.1073/pnas.1207968109/-/DCSupplemental>.
11. Maxson ME, Naj X, O'meara TR, Plumb JD, Cowen LE, Grinstein S. 2018. Integrin-based diffusion barrier separates membrane domains enabling the formation of microbiostatic frustrated phagosomes <https://doi.org/10.7554/eLife.34798.001>.
12. Domingues N, Catarino S, Cristovao B, Rodrigues L, Filomena C, Sarmiento MJ, Almeida J, Rodrigues-Santos P, Korolchuk V, Gonçalves T, Raimundo N. 2022. Cx43 promotes exocytosis of damaged lysosomes through actin remodelling <https://doi.org/10.21203/RS.3.RS-2277227/V1>.

July 27, 2023

Prof. Henrique Girao
University of Coimbra, Faculty of Medicine, Coimbra Institute for Clinical and Biomedical Research
iCBR
Polo III - Health Sciences Campus • Azinhaga Santa Comba, Celas | Portugal
Coimbra 3000-548
Portugal

Re: Spectrum01238-23R1 (**Cx43-mediated hyphal folding counteracts phagosome integrity loss during fungal infection**)

Dear Prof. Henrique Girao:

The reviewers were happy with the revisions. They commented that the changes were made in a thoughtful and comprehensive manner. Your manuscript is now acceptable for publication.

Your manuscript has been accepted, and I am forwarding it to the ASM Journals Department for publication. You will be notified when your proofs are ready to be viewed.

Sincerely,

James Konopka
Editor, Microbiology Spectrum
